# Online Learning in Risk Sensitive constrained MDP

**Arnob Ghosh** [* 1]   **Mehrdad Moharrami** [* 2]

## Abstract

We consider a setting in which the agent aims to maximize the expected cumulative reward, subject to a constraint that the entropic risk of the total utility exceeds a given threshold. Unlike the risk-neutral case, standard primal-dual approaches fail to directly yield regret and violation bounds, as value iteration with respect to a combined state-action value function is not applicable in the risk-sensitive setting. To address this, we adopt the Optimized Certainty Equivalent (OCE) representation of the entropic risk measure and reformulate the problem by augmenting the state space with a continuous budget variable. We then propose a primal-dual algorithm tailored to this augmented formulation. In contrast to the standard approach for risk-neutral CMDPs, our method incorporates a truncated dual update to account for the possible absence of strong duality. We show that the proposed algorithm achieves regret of $\tilde{\mathcal{O}}\big(V_{g,\max}K^{3/4} + \sqrt{H^4 S^2 A \log(1/\delta)}K^{3/4}\big)$ and constraint violation of $\tilde{\mathcal{O}}\big(V_{g,\max}\sqrt{H^3 S^2 A \log(1/\delta)}K^{3/4}\big)$ with probability at least $1 - \delta$, where $S$ and $A$ denote the cardinalities of the state and action spaces, respectively, $H$ is the episode length, $K$ is the number of episodes, $\alpha < 0$ is the risk-aversion parameter, and $V_{g,\max} = \frac{1}{|\alpha|}(\exp(|\alpha|H) - 1)$. To the best of our knowledge, this is the first result establishing sublinear regret and violation bounds for the risk-sensitive CMDP problem.

## 1. Introduction

Reinforcement Learning (RL) has achieved remarkable breakthroughs across a wide range of domains, in-cluding human-level performance on classic Atari 2600 games (Mnih et al., 2015), mastering complex board games such as Go and Chess (Silver et al., 2016; 2017), tackling challenging robotic manipulation tasks (Levine et al., 2016; Haarnoja et al., 2018), optimizing control systems for autonomous vehicles (Sallab et al., 2017), advancing drug discovery by designing molecules with desired properties (Popova et al., 2018), and enhancing the capabilities of Large Language Models to achieve human-level decision-making in natural language understanding and generation tasks (Ouyang et al., 2022; Guo et al., 2025). These successes underscore the versatility of RL in high-dimensional, sequential decision-making problems. However, most of these methods focus on maximizing a single reward signal without regard to auxiliary performance or safety constraints, which are often critical in real-world applications.

Constrained Markov Decision Processes (CMDPs) provide a principled way to incorporate such additional constraints into the decision-making process (Altman, 1999b). By extending the standard RL objective to include one or more cost functions, CMDPs enable practitioners to balance primary performance goals against other considerations such as safety (García & Fernández, 2015), resource allocation (Zhao et al., 2022), or fairness (Jabbari et al., 2017). Various techniques have been proposed to solve CMDPs, including primal-dual methods (Efroni et al., 2020) and Lagrangian relaxation (Altman, 1999b; Borkar, 2005).

More recently, it has been shown that CMDPs has zero duality gap under certain conditions (Paternain et al., 2019), and this result has been leveraged in a series of papers to derive regret bounds for model-free CMDPs (Ghosh et al., 2022; Ding et al., 2021). While these methods are effective for constraints that are linear in cost (e.g., requiring that the expected sum of certain costs remains below a threshold), many real-world tasks demand handling nonlinear, risk-sensitive constraints that capture higher-order statistics or worst-case scenarios (Chow et al., 2018). Such constraints are paramount in domains where even a single catastrophic event is unacceptable, such as in healthcare (Gottesman et al., 2019). However, incorporating risk-sensitive constraints into CMDPs introduces significant challenges, as the resulting problem often becomes highly nonlinear, and the strong duality properties that simplify solution methods for CMDPs may fail to hold. Motivated by this, we are

---

*Equal contribution   [1]Department of Electrical and Computer Engineering, New Jersey Institute of Technology, Newark, USA [2]Department of Computer Science, University of Iowa, Iowa City, USA. Correspondence to: Arnob Ghosh <arnob.ghosh@njit.edu>.

*Proceedings of the $42^{nd}$ International Conference on Machine Learning*, Vancouver, Canada. PMLR 267, 2025. Copyright 2025 by the author(s).

interested in the following question:

*Can we achieve a provably efficient learning algorithm for CMDP problem with risk-sensitive constraints?*

In this work, we study a finite-horizon CMDP with an entropic risk-sensitive constraint. In particular, we consider that the agent seeks to maximize the expected cumulative reward subject to the constraint that the risk-sensitive utility associated with a policy remains above a prescribed threshold in an online learning setup. Solving this problem exactly is notoriously difficult due to its nonlinear nature and the potential failure of strong duality.

**Our Contributions**:

- We show that the proposed algorithm, with probability at least $1 - \delta$, achieves a regret bound of $\tilde{\mathcal{O}}\big(V_{g,\max}K^{3/4} + \sqrt{H^4S^2A\log(1/\delta)}K^{3/4}\big)$ and a constraint violation bound of $\tilde{\mathcal{O}}\big(V_{g,\max}\sqrt{H^3S^2A\log(1/\delta)}K^{3/4}\big)$, where $S$ and $A$ denote the cardinalities of the state and action spaces, $H$ the episode length, $K$ the number of episodes, $\alpha < 0$ the risk-aversion parameter, and $V_{g,\max} = \frac{1}{|\alpha|}(\exp(|\alpha|H)-1)$. To the best of our knowledge, this is the first result establishing sublinear regret and violation bounds for the risk-sensitive CMDP problem.

- The standard primal-dual approach may not apply in this setting, as the exponential Bellman equation for the composite state-action value function is invalid due to the non-linearity of the entropic risk measure where markovian policy is optimal. Moreover, we show that a Markovian policy might be suboptimal unlike the unconstraned entropic risk measure setting. To address this, we introduce an augmented state space by extending the original state-space with a continuous budget variable. We show that, in the risk-averse case ($\alpha < 0$), the entropic risk associated with a policy $\pi$ can be expressed as $\max_\tau (\tau + \mathbb{E}[u(G(\pi) - \tau)])$, where $u(t) = \frac{1}{\alpha}(\exp(\alpha t) - 1)$ and $G(\pi)$ denotes the total utility under policy $\pi$. By reformulating the problem in the augmented state space, we can apply a value-iteration-based approach to compute the composite state-action value function.

- We then apply a primal-dual-based approach. However, since the entropic risk measure is not linear in the state-action occupancy measure, strong duality may not hold. As a result, standard techniques used in the risk-neutral CMDP setting for bounding constraint violation (Ghosh et al., 2022; Efroni et al., 2020; Ding et al., 2021) are not directly applicable. To address this, we introduce a truncation variable $\xi$ and show that it is crucial for controlling both regret and constraint violation.

## 1.1. Related Works

In this section, we provide a brief overview of the most relevant related work. Specifically, we focus on risk-sensitive MDPs and constrained MDPs, as the use of entropic risk measures in constrained MDPs remains largely unexplored.

**Risk Sensitive RL:** The study of risk sensitive RL traces its origins to economic theory, where risk was incorporated into utility functions to better capture decision-making under uncertainty (Hardy, 1923; von Neumann & Morgenstern, 1944; Luce & Raiffa, 1957; Arrow, 1965). Risk sensitive Markov decision processes were first introduced in (Howard & Matheson, 1972), where the standard cost function was replaced by an exponential transformation involving a risk sensitive parameter. This framework is closely aligned with the concepts of robustness (Dai Pra et al., 1996; Hernandez-Hernández & Marcus, 1996), as it balances utility maximization while accounting for the variance of returns (Whittle, 2002). The use of exponential utility functions effectively penalizes variability in outcomes, thereby mitigating risk in decision-making.

Risk sensitivity entered the control and reinforcement learning literature in the early 2000s (Borkar, 2001; Borkar & Meyn, 2002; Borkar, 2002). Since then, significant efforts have been made to understand the behavior of exponential risk sensitive reinforcement learning, with notable recent contributions including (Fei et al., 2020; Murthy et al., 2023; Fei et al., 2024; Moharrami et al., 2024). Simultaneously, alternative risk measures such as Conditional Value-at-Risk (CVaR) gained traction (Artzner et al., 1999; Rockafellar & Uryasev, 2000; Wang et al., 2023; 2024). CVaR has become particularly valuable in reinforcement learning applications, as it focuses on optimizing expected rewards in worst case scenarios (Tamar et al., 2015; Zhao et al., 2024). In this work, we focus on solving the problem where the objective is to maximize the cumulative reward subject to the constraint the entropic risk measure associated with the utility value function is above a certain threshold. Hence, it is fundamentally different from the above work.

**Constrained RL:** Constrained RL is one of the most active fields at the intersection of constraint optimization, MDPs, and RL. The earliest attempts to integrate constraints into MDP formulations date back to the 1970s (Kolesar, 1970), with more rigorous treatments appearing in the 1990s (Ross, 1989; Altman, 1999a). Optimization techniques for solving constraint problems, such as Lagrangian relaxation (Everett, 1963; Shapiro, 1979), and primal-dual (Efroni et al., 2020) approach have been used to address these challenges, often by converting the constrained problem into an unconstrained one through the use of Lagrange multipliers (Altman, 1998; Bertsekas, 2016). This framework was later adapted to RL (Geibel & Wysotzki, 2005; Uchibe & Doya, 2007; Zheng & Ratliff, 2020; Tessler et al., 2019; Ding

et al., 2020; Ying et al., 2022). Since then, numerous studies have explored different formulations of constrained objectives within the RL framework, including constraints with auxiliary cost/reward functions (Achiam et al., 2017; Tessler et al., 2019; Ghosh et al., 2022), constraints on the quantile and distribution of returns (Jung et al., 2024), and constraints on risk measures (Borkar & Jain, 2014; Zhang et al., 2024). Compared to the risk-neutral constraints, we consider risk-sensitive constraints. Furthermore, to the best of our knowledge, this is the first work that provides the sub-linear regret and violation bound for risk-sensitive constraints setup.

## 2. Problem Formulation

We consider the risk-sensitive CMDP within an episodic framework. The CMDP is defined as $(\mathcal{S}, \mathcal{A}, \mathbb{P}, H, r, g)$, where $\mathcal{S}$ ($|\mathcal{S}| = S$) is the finite state space, $\mathcal{A}$ ($|\mathcal{A}| = A$) is the finite action space, and $H$ is the fixed episode length. The transition dynamics are governed by a set of transition probabilities $\mathbb{P} = \{\mathbb{P}_h\}_{h=1}^{H}$, where $\mathbb{P}_h(s' \mid s, a)$ represents the probability of transitioning to state $s'$ from state $s$ at time step $h$, upon taking action $a$. The reward and utility functions are denoted by $r = \{r_h\}_{h=1}^{H}$ and $g = \{g_h\}_{h=1}^{H}$, respectively, defined for each time step of the episode and assumed to be deterministic and in $[0, 1]$.

Each episode $k \geq 0$ starts at a fixed state $s_1$. At each time step $h$, the agent observes the state $s_h^k \in \mathcal{S}$ and selects an action $a_h^k \in \mathcal{A}$. The agent then receives a reward $r_h(s_h^k, a_h^k)$ and a utility $g_h(s_h^k, a_h^k)$. Finally, the MDP evolves to $s_{h+1}^k$ drawn from $\mathbb{P}_h(\cdot|s_h^k, a_h^k)$. The episode terminates at step $H + 1$. Without loss of generality, we assume that $r_{H+1} = g_{H+1} = 0$. In this paper, we consider the challenging scenario where the agent only observes $r_h(s_h^k, a_h^k)$ and $g_h(s_h^k, a_h^k)$ at the visited state-action pairs.

The agent's history-dependent policy space is denoted by

$$\Pi_{\mathrm{HD}} = \Big\{ \pi = \{\pi_h(\cdot \mid \cdot)\}_{h=1}^{H} :$$

$$\pi_h(\cdot \mid s_h, \{s_{h'}, a_{h'}\}_{h'=1}^{h-1}) \in \Delta(\mathcal{A}), \ \forall h \in [H] \Big\},$$

where $\Delta(\mathcal{A})$ denotes the probability simplex over the action space. We consider a total of $K$ episodes. For each $k \leq K$ and $h \leq H$, let $\pi_h^k$ denote the policy at time step $h$ of episode $k$.

The reward value function is the expected total reward $V_{r,1}^{\pi}(s) = \mathbb{E}_{\pi}[\sum_{h=1}^{H+1} r_h(s_h, a_h)|s_1 = s]$. The constraint is based on the risk associated with the utility function. For risk-sensitive RL with the risk factor $\alpha$, we denote the risk-sensitive state-action value function for the utility as $Q_{g,1}^{\pi} : \mathcal{S} \times \mathcal{A} \to \mathbb{R}$. This function represents the expected cumulative utility, assessed using the entropic risk measure,

under a policy $\pi \in \Pi_{\mathrm{HD}}$, starting from the state-action pair $(s, a)$, i.e.,

$$Q_{g,1}^{\pi}(s, a) = \frac{1}{\alpha} \log \left( \mathbb{E}_{\pi} \left[ e^{\alpha \sum_{h'=1}^{H+1} g_{h'}(s_{h'}, a_{h'})} \Big| s_1 = s, a_1 = a \right] \right).$$

Similarly, the risk-sensitive value function for the utility associated with policy $\pi$, starting from state $s$ is given by

$$V_{g,1}^{\pi}(s) = \frac{1}{\alpha} \log \left\{ \mathbb{E}_{\pi}[e^{\alpha \sum_{h'=1}^{H+1} g_{h'}(s_{h'}, a_{h'})}|s_1 = s] \right\}$$

For a Markov policy $\pi$, the state-action value function and the utility-based value function satisfy the following dynamic programming relations:

$$Q_{g,h}^{\pi}(s, a) = g_h(s, a) + \frac{1}{\alpha} \log \left( \mathbb{P}_h e^{\alpha V_{g,h+1}^{\pi}}(s, a) \right),$$

$$V_{g,h}^{\pi}(s) = \frac{1}{\alpha} \log \left( \langle e^{\alpha Q_{g,h}^{\pi}(s, \cdot)}, \pi(\cdot|s) \rangle \right).$$

Our objective is to solve the following risk-sensitive constrained problem:

$$\max_{\pi \in \Pi_{HD}} V_{r,1}^{\pi}(s_1), \qquad \text{subject to } V_{g,1}^{\pi}(s_1) \geq \mathcal{B}, \quad (1)$$

where $\mathcal{B} \in \mathbb{R}$ is the lower bound on acceptable utility. The goal is to ensure that the entropic risk measure associated with the utility value function is above a certain threshold. We consider the setting where $\alpha < 0$, indicating that the agent is risk-averse. Here, $\alpha$ denotes the agent's *risk tolerance*. As $\alpha \uparrow 0$, the agent approaches risk-neutral behavior and seeks policies whose expected cumulative utility is at least $\mathcal{B}$. Conversely, as $\alpha \downarrow -\infty$, the agent becomes increasingly risk-averse and favors policies that yield outcomes with lower volatility and a greater certainty of meeting or exceeding the threshold $\mathcal{B}$. Problems of this type are essential in various applications, including autonomous driving, financial investment, and modeling human behavior.

**Learning Metric**: In this setting, we assume the agent does not have access to the transition probabilities and must learn in an online manner. The agent aims to minimize the following performance metrics:

$$\mathrm{Regret}(K) = \sum_{k=1}^{K} V_{r,1}^{\pi^*}(s_1) - V_{r,1}^{\pi^k}(s_1),$$

$$\mathrm{Violation}(K) = \sum_{k=1}^{K} (\mathcal{B} - V_{g,1}^{\pi^k}(s_1))$$

where $\pi^k \in \Pi_{\mathrm{HD}}$ is the policy deployed at episode $k$, and $\pi^*$ represents the optimal feasible policy. The regret quantifies the sub-optimality gap, while the violation measures the extent of constraint violations.

# 3. Solution Methodology

We begin by highlighting the challenges in solving the risk-sensitive CMDP compared to the standard CMDP. In the standard CMDP, which corresponds to the risk-neutral scenario ($\alpha = 0$), the Lagrangian is solved to derive a bound on the learning metric (Ghosh et al., 2022):

$$\min_{\lambda \geq 0} \max_{\pi} V_{r,1}^{\pi}(x) + \lambda(V_{g,1}^{\pi}(x) - b) \tag{2}$$

**Challenge in applying value iteration:** In the standard CMDP setting, for a fixed $\lambda$, the optimal Markov policy $\pi$ can be obtained using a standard dynamic programming approach as the problem reduces to an unconstrained MDP with a modified per-step reward of $r + \lambda g$. Hence, a value-based method can be applied to compute the composite value function. This approach is then used to iteratively update both the policy $\pi$ and the dual variable $\lambda$.

However, in the risk-sensitive setting, expanding the Lagrangian yields:

$$r_h(x, a) + \mathbb{P}_h V_{r,h+1}^{\pi}(x, a) + \lambda g_h(x, a) +$$
$$\frac{\lambda}{\alpha} \log \left( \mathbb{P}_h e^{\alpha V_{g,h+1}^{\pi}(s, a)} \right)$$

which prevents the use of standard dynamic programming to solve for the optimal policy at a fixed $\lambda$. Unlike the risk-neutral case, the Lagrangian here does not reduce to a problem with a simple modified reward $r + \lambda g$.

**Lack of strong duality**: For the standard CMDP, it has been shown that strong duality holds if a strict feasible policy (a.k.a. Slater's condition) exists (Paternain et al., 2019) exists. Thus, one can solve in the dual-domain. For standard CMDP problems, (Ghosh et al., 2022) demonstrate that primal-dual based approaches can achieve a $\mathcal{O}(\sqrt{K})$ regret and violation using the strong duality result. The key to obtain strong duality result in (Paternain et al., 2019) is that the value function is linear in the state-action occupancy measure. However, the risk-sensitive state-action value function is not linear with respect to the occupancy measure, and the risk-sensitive CMDPs may not satisfy strong duality. Consequently, traditional primal-dual-based algorithms are inadequate for obtaining the desired results.

**Absence of a Markov Optimal Policy**: In contrast to standard CMDPs, the lack of strong duality in the risk-sensitive setting means that one cannot guarantee the existence of a Markov policy that solves the CMDP. The existence of such a policy is crucial for deriving finite-time bounds. In Appendix A, we present an example demonstrating that the optimal policy for a CMDP with a entropic risk constraint may not be even Markovian.

## 3.1. Certainty-Equivalence Representation

To address these issues, we leverage the certainty-equivalence representation of the entropic risk measure (Ben-Tal & Teboulle, 2007). Specifically, for a function $u(\cdot)$, the Optimized Certainty Equivalent (OCE) associated with a $\pi \in \Pi_{HD}$ is defined as $\mathrm{OCE}_u(\pi, s) = \max_\tau \left\{ \tau + \mathbb{E}\left[ u\left( \sum_{h=1}^{H} g_h(s_h, a_h) - \tau \right) \mid s_1 = s \right] \right\}$

In the risk-averse case ($\alpha < 0$), the entropic risk measure can be recovered by setting $u(t) = \frac{1}{\alpha}(e^{\alpha t} - 1)$. For notational convenience, we use $\mathrm{OCE}(\pi)$ from this point onward, as the function $u(\cdot)$ and the initial state $s_1$ is fixed.

**Lemma 3.1.** *For $\alpha < 0$, we have $\mathrm{OCE}(\pi) = V_{g,1}^{\pi}(s_1)$.*

Unlike the entropic risk measure, the OCE does not admit a dynamic programming formulation. As a result, the optimal policy that maximizes $\mathrm{OCE}(\pi)$ is generally history-dependent. The solution is to consider an augmented MDP that extends the state space by introducing a scalar budget variable, which intuitively tracks the cumulative utility over time. This technique enables a reformulation of the problem into a state-based representation that is amenable to dynamic programming. This approach has been employed in the context of CVaR and spectral risk measures, and has more recently extended to the OCE framework (Bäuerle & Ott, 2011; Wang et al., 2024). We build on this formulation and extend it to our CMDP state-space.

## 3.2. Augmented CMDP

Consider an augmented CMDP $(\mathcal{S}^{\mathrm{aug}}, \mathcal{A}, \mathbb{P}^{\mathrm{aug}}, H, r, g^{\mathrm{aug}})$ by appending a budget variable to the state and modifying the underlying utility function (Bäuerle & Ott, 2011; Wang et al., 2024). More specifically, we augment the state space with a budget variable $c_h$ at horizon $h$ defined by $c_h = \tau - \sum_{h'=1}^{h-1} g_{h'}(s_{h'}, a_{h'})$, where $c_1 = \tau \in [0, H]$ is the initial budget provided by the algorithm, and $c_h$ denotes the remaining budget at time step $h$. Note that $c_h \in [-H, H]$.

The budget transition is known and deterministic: $c_{h+1} = c_h - g_h(s_h, a_h)$. We define the augmented utility function $g_h^{\mathrm{aug}}$ by $g_h^{\mathrm{aug}}(s, c, a) = 0$ for $h \leq H$, and $g_{H+1}^{\mathrm{aug}}(s, c, a) = u(-c)$, where $u(t) = \frac{1}{\alpha}(e^{\alpha t} - 1)$. The choice of $u(t)$ is dictated by the entropic risk measure. Note that the transition probability for the augmented CMDP problem is given by $\mathbb{P}_h^{\mathrm{aug}}(\cdot, c'|s, c, a) = \mathbb{P}_h(\cdot|s, a)$ for $c' = c - g_h(s, a)$, and $\mathbb{P}_h^{\mathrm{aug}}(\cdot, c'|s, c, a) = 0$, otherwise, as $g_h$ is deterministic.

The agent focuses on Markov policies defined over the augmented state space, denoted by

$$\Pi_{\mathrm{M}}^{\mathrm{aug}} = \left\{ \pi = \{\pi_h(\cdot \mid \cdot)\}_{h=1}^{H} : \pi_h(\cdot \mid s_h, c_h) \in \Delta(\mathcal{A}), \right.$$

$$\left. \forall h \in [H] \text{ and } c \in [-H, H] \right\}.$$

For a Markov policy $\pi$ in the augmented state space, abusing the notation, let $Q_{g,h}^\pi$ and $V_{g,h}^\pi$ denote the augmented state-action value function and the augmented state-value function, respectively. The distinction between the risk-sensitive MDP and the augmented MDP is evident from the difference in the size of their state spaces. By definition,

$$Q_{g,h}^\pi(s,c,a) =$$
$$\mathbb{E}\Big[\sum_{h'=h}^{H+1} g_{h'}^{\text{aug}}(s_{h'},c_{h'},a_{h'})\big|s_h=s,c_h=c,a_h=a\Big],$$

$$V_{g,h}^\pi(s_h,c_h) = \mathbb{E}\Big[\sum_{h'=h}^{H+1} g_{h'}^{\text{aug}}(s_{h'},c_{h'},a_{h'})\big|s_h=s,c_h=c\Big].$$

Note that for a Markov policy $\pi$ in the augmented state space, the augmented CMDP admits a linear state-action value function with respect to the utility function $g_h^{\text{aug}}$, in contrast to the risk-sensitive value functions $Q_{g,h}^\pi(s,a)$ and $V_{g,h}^\pi(s)$ defined for a Markov policy $\pi$ in the original state space $\mathcal{S}$. This is achieved by shifting the utility to the terminal time step, i.e., $V_{g,H+1}^\pi(s_H,c_H) = u(-c_{H+1})$. The resulting additive structure is the key property that enables the analysis of Lagrangian relaxation.

Furthermore, the function $V_{g,h}(s,\cdot)$ is bounded above by $V_{g,\max} = \frac{1}{|\alpha|}(\exp(|\alpha|H) - 1)$. Finally, for a Markov policy $\pi$ in the augmented MDP, the functions $Q_{g,h}^\pi$ and $V_{g,h}^\pi$ satisfy standard dynamic programming equations:

$$Q_{g,h}^\pi(s_h,c_h,a_h) = \mathbb{E}\big[V_{g,h+1}^\pi(s_{h+1},c_{h+1}) \mid s_h,c_h,a_h\big],$$
$$V_{g,h}^\pi(s_h,c_h) = \sum_{a\in\mathcal{A}} \pi(a \mid s_h,c_h)Q_{g,h}^\pi(s_h,c_h,a).$$

Note that OCE selects the optimal trade-off between the initial budget and the value function of the augmented CMDP: $\text{OCE}(\pi) = \max_\tau \big(\tau + V_{g,1}^\pi(s_1,\tau)\big)$. Given this relationship, we present the augmented risk-sensitive CMDP formulation, which is equivalent to the original problem (1). Since policies are defined over the augmented state space, we denote the reward value function as $V_{r,1}^\pi(s_1,\tau)$.

$$\max_\pi V_{r,1}^\pi(s_1,\hat{\tau})$$
$$\text{subject to } \hat{\tau} = \arg\max\{(\tau + V_{g,1}^\pi(s_1,\tau))\} \quad (3)$$
$$(\hat{\tau} + V_{g,1}^\pi(s_1,\hat{\tau})) \geq \mathcal{B}.$$

(Wang et al., 2024) studied an augmented formulation in the unconstrained setting for various risk measures. However, to the best of our knowledge, our work is the first to extend the augmented CMDP framework to the constrained setting. Unlike the unconstrained case, the constrained setting requires controlling both regret and constraint violation. We make the following assumption, which, in the risk-neutral CMDP setting, follows by Slater's condition (Paternain et al., 2019). Deriving an analogous condition for the augmented CMDP is left for future work.

**Assumption 3.2.** There exists a Markov policy in the augmented state space that solves the augmented CMDP.

Note that a complete history-dependent policy will be computationally more challenging, hence, we focus on Markovian policy class on the augmented state space.

## 4. Algorithm

Considering a Lagrangian relaxation of (3), our goal is to solve the following optimization problem:

$$\min_{\lambda\geq 0} \max_\pi \max_\tau V_{r,1}^\pi(s_1,\tau) + \lambda\big((\tau + V_{g,1}^\pi(s_1,\tau)) - \mathcal{B}\big). \quad (4)$$

Note that the order of maximization in the above formulation is interchangeable. Since the augmented value function $V_{g,1}^\pi(s_1,\tau)$ satisfies a linear Bellman equation, the inner maximization, for a fixed $\tau$ and $\lambda \geq 0$, can be efficiently solved using standard value iteration algorithms. In particular, for a fixed $\tau$ and $\lambda \geq 0$, one can find the optimal policy that maximizes the composite state-action value function.

The above approach addresses the primary challenge faced by the original risk-sensitive CMDP. However, the absence of strong duality remains a fundamental challenge. Specifically, the key issue is determining an appropriate tuning strategy for $\lambda$ to ensure a bounded trade-off between policy regret and constraint violation.

A natural approach to updating the Lagrange multiplier $\lambda \geq 0$ is to use gradient descent. However, due to the lack of strong duality, the argument from (Ghosh et al., 2022) cannot be directly applied to establish its boundedness. In particular, the value function is still non-convex in terms of the augmented state-action occupancy measure. Hence, one can not apply the tool used in (Paternain et al., 2019) to prove strong duality. To address this issue, we consider a truncated dual update, capped at a threshold $\xi$ to be specified later. This truncation is crucial for achieving both the regret and constraint violation bounds.

**Algorithm Description**: We now describe our proposed Algorithm 1. The algorithm maintains an empirical estimate of the transition probabilities, which is updated as new data is observed (line 6). At each episode, this estimate is then used to compute an optimistic estimate of the reward state-action value function and the utility state-action value function, using a bonus term for every augmented state-action pair (lines 8–11). This bonus term enforces exploration.

Next, the policy is updated via a greedy algorithm with respect to the estimated composite state-action value function given by the Lagrangian relaxation (line 13). We then update the reward and utility value functions using standard dynamic programming equations (line 14), which effectively solves the inner maximization of the Lagrangian relaxation for any budget $c$ in the discretized budget space $\mathcal{C}$ and a fixed

$\lambda_k > 0$. Subsequently, we obtain the optimal initial budget $\hat{\tau}_k$ using the current estimate of $V_{g,1}^{\pi^k}(s_1, \hat{\tau})$ and $V_{r,1}^{\pi^k}(s_1, \hat{\tau})$ to solve the inner maximization of (4) (line 16). Finally, we tune the dual variable via a gradient-descent update on $\lambda$:

$$\lambda_{k+1} = \text{Proj}_{[0,\xi]}\left(\lambda_k + \eta(\mathcal{B} - (\hat{\tau}_k + V_{g,1}(s_1, \hat{\tau}_k)))\right)$$

where $\text{Proj}_{[0,\xi]}$ denotes the projection operator onto the interval $[0, \xi]$. In line 19, we execute the updated policy $\pi_h^k$ using the optimized $\hat{\tau}_k$ as the initial budget. We track the evolution of the budget while executing the policy.

**Computational Challenge**: One key challenge in optimizing $\tau$ is that the objective function is not concave in $\tau$. To address this, we discretize the budget variable with resolution $\epsilon_0$, where $\epsilon_0$ is chosen to ensure polynomial-time complexity while preserving regret and violation guarantees. We denote the resulting discretized set of feasible budget values by $\mathcal{C}$. Note that $\mathcal{C} \subseteq [-H, H]$ and that $|\mathcal{C}| = \lceil 2H/\epsilon_0 \rceil$.

# 5. Analysis

## 5.1. Main Result

We now present our main theoretical contribution, which establishes high-probability sublinear bounds on both regret and constraint violation for Algorithm 1. Our analysis extends existing frameworks in risk-sensitive and constrained reinforcement learning.

**Theorem 5.1.** *With probability $1 - \delta$, Algorithm 1 returns the policies $\{\pi^k\}_{k=1}^K$ such that*

$$\text{Regret}(K) \leq \tilde{\mathcal{O}}\left(V_{g,\max}K^{3/4} + \sqrt{H^4 S^2 A \log(1/\delta)}K^{3/4}\right),$$

$$\text{Violation}(K) \leq \tilde{\mathcal{O}}\left(V_{g,\max}K^{3/4}\sqrt{H^3 S^2 A \log(1/\delta)}\right),$$

*where $V_{g,\max} = \frac{1}{|\alpha|}(\exp(|\alpha|H) - 1)$, and $\tilde{\mathcal{O}}(\cdot)$ hides logarithmic factors in $S$, $A$, $H$, and $K$.*

This is the first sublinear regret and violation result for the risk-sensitive constrained MDP setting. In the unconstrained, risk-neutral setting, the best known regret bound is $\tilde{\mathcal{O}}(\sqrt{H^3 SAK \log(1/\delta)})$ (Azar et al., 2017). The additional $\sqrt{HS}$ factor in our bound arises from the use of an augmented state space; specifically, we must apply uniform, value-aware concentration bounds over the discretized augmented state space. The additional $K^{1/4}$ factor arises from carefully balancing the discretization error with the size of the discretized budget set $\mathcal{C}$.

For the unconstrained risk-sensitive MDP setting, the regret bound is $\mathcal{O}(V_{g,\max}\sqrt{H^2 S^2 AK \log(1/\delta)})$, which is known to be tight (Fei et al., 2021). In our constrained setting, we achieve a violation bound of $\tilde{\mathcal{O}}(V_{g,\max}\sqrt{H^3 S^2 A \log(1/\delta)}K^{3/4})$. The additional

$\sqrt{H}K^{1/4}$ factor arises from the use of a discretized augmented state space and the introduction of the truncation threshold $\xi$, as strong duality may not hold.

## 5.2. Proof Outline

We first derive the regret and constraint violation associated with the problem formulation in the augmented CMDP. We then discuss the relationship between the regret and violation in the augmented state space and those in the original CMDP. Note that the policy $\pi^k$ depends on the discretized initial budget $\hat{\tau}_k$ and the discretized utilities observed:

$$
\begin{aligned}
\text{Regret}^{\text{aug}}(K) &= \sum_k (V_{r,1}^{\pi^*}(s_1, \tau_*) - V_{r,1}^{\pi^k}(s_1, \hat{\tau}_k)), \\
\text{Violation}^{\text{aug}}(K) &= \sum_k (\mathcal{B} - (\hat{\tau}_k + V_{g,1}^{\pi^k}(s_1, \hat{\tau}_k))),
\end{aligned}
\tag{5}
$$

where $\pi^*$ is the optimal feasible policy for the original risk-sensitive CMDP, $\hat{\tau}_k$ is given by Algorithm 1, and $\tau_* = \arg\max_\tau(\tau + V_{g,1}^{\pi^*}(s_1, \tau))$. Note that the value of $\hat{\tau}_k$ determines the policy $\pi^k$.

**Step 1**: Let $V_{g,1}^k$ and $V_{r,1}^k$ denote the optimistic estimates of $V_{g,1}^{\pi^k}$ and $V_{r,1}^{\pi^k}$, respectively, as computed by Algorithm 1 on line 12. We begin with the following observation:

$$
\begin{aligned}
&\sum_k \left(V_{r,1}^{\pi^*}(x, \tau_*) - V_{r,1}^{\pi^k}(x, \hat{\tau}_k)) + \lambda(\mathcal{B} - (\hat{\tau}_k + V_{g,1}^{\pi^k}(x, \hat{\tau}_k))\right) \\
&\leq \underbrace{\sum_k (V_{r,1}^k(x, \hat{\tau}_k) - V_{r,1}^{\pi^k}(x, \hat{\tau}_k))}_{\mathcal{T}_1} \\
&\quad + \underbrace{\sum_k (V_{r,1}^{\pi^*}(x, \tau_*) - V_{r,1}^k(x, \hat{\tau}_k)) +}_{\substack{\cdots \\ \lambda_k((\tau_* + V_{g,1}^{\pi^*}(x, \tau_*)) - (\hat{\tau}_k + V_{g,1}^k(x, \hat{\tau}_k))) \\ \cdots}} \\
&\hspace{5em} \mathcal{T}_2 \\
&\quad + \underbrace{\sum_k (\lambda - \lambda_k)(\mathcal{B} - (\hat{\tau}_k + V_{g,1}^k(x, \hat{\tau}_k)))}_{\mathcal{T}_3} \\
&\quad + \lambda \underbrace{\sum_k ((\hat{\tau}_k + V_{g,1}^k(x, \hat{\tau}_k)) - (\hat{\tau}_k + V_{g,1}^{\pi^k}(x, \hat{\tau}_k)))}_{\mathcal{T}_4}
\end{aligned}
$$

where the inequality follows from the fact that $(\tau_* + V_{g,1}^{\pi^*})(x) \geq \mathcal{B}$. Note that when $\lambda = 0$, the above expression reduces to the regret. This same expression also suffices for bounding the violation, as we explain later.

**Step 2**: We bound $\mathcal{T}_3$ by analyzing the truncated dual update for constraint violation.

**Algorithm 1** Constraint Risk Sensitive Value Iteration Algorithm

---

**Input:** Number of episodes $K \in \mathbb{Z}^+$, discretized budget space $\mathcal{C}$, confidence level $\delta \in (0,1]$, risk parameter $\alpha < 0$, utility lower bound $\mathcal{B}$, learning rate $\eta$, initial policy $\pi^0$, truncation threshold $\xi$, and $V_{g,\max} = \frac{1}{|\alpha|}(\exp(|\alpha|H) - 1)$.

1: $\lambda_1 \leftarrow 0$
2: For all $(s, \hat{c}, a) \in \mathcal{S} \times \mathcal{C} \times \mathcal{A}$, initialize $V_{g,H+1}(s, \hat{c}) \leftarrow u(-\hat{c})$ and $V_{r,H+1}(s, \hat{c}) \leftarrow 0$
3: For all $(h, s, \hat{c}, a) \in [H] \times \mathcal{S} \times \mathcal{C} \times \mathcal{A}$, set $Q_{r,h}(s, \hat{c}, a) \leftarrow H$, $Q_{g,h}(s, \hat{c}, a) \leftarrow V_{g,\max}$
4: For all $h \in [H]$, initialize dataset $\mathcal{D}_h \leftarrow \emptyset$
5: **for** episode $k = 1$ to $K$ **do**
6:     For each $h \in [H]$, compute counts and empirical transitions: $N_h(s, a, s') \leftarrow \sum_{i=1}^{k-1} \mathbb{1}[(s_h^i, a_h^i, s_{h+1}^i) = (s, a, s')]$,
    $N_h(s, a) \leftarrow \max\{1, \sum_{s'} N_h(s, a, s')\}$, and $P_h^k(s' \mid s, a) \leftarrow \frac{N_h(s,a,s')}{N_h(s,a)}$
7:     **for** step $h = H$ to 1 **do**
8:         **for** all $(s, \hat{c}, a) \in \mathcal{S} \times \mathcal{C} \times \mathcal{A}$ **do**
9:         
$$Bon_{r,h}(s,a) \leftarrow 9H\sqrt{\frac{S|\mathcal{C}|\log(HSAK|\mathcal{C}|/\delta)}{N_h(s,a)}}, \; Bon_{g,h}(s,a) \leftarrow 6V_{g,\max}\sqrt{\frac{S|\mathcal{C}|\log(HSAK|\mathcal{C}|V_{g,\max}/\delta)}{N_h(s,a)}}$$
10:         
$$Q_{g,h}(s,\hat{c},a) \leftarrow \min\left\{\mathbb{E}_{s'\sim P_h^k(\cdot|s,a)}\left[V_{g,h+1}(s', \hat{c} - \phi(g_h(s,a)))\right] + Bon_{g,h}(s,a), V_{g,\max}\right\}$$
11:         
$$Q_{r,h}(s,\hat{c},a) \leftarrow \min\left\{r_h(s,a) + \mathbb{E}_{s'\sim P_h^k(\cdot|s,a)}\left[V_{r,h+1}(s', \hat{c} - \phi(g_h(s,a)))\right] + Bon_{r,h}(s,a), H\right\}$$
12:         **end for**
13:         For all $(s, \hat{c}) \in \mathcal{S} \times \mathcal{C}$, set $\pi_h^k(a \mid s, \hat{c}) = 1$ for some $a \in \arg\max_{a' \in \mathcal{A}}(Q_{r,h}(s, \hat{c}, a') + \lambda_k Q_{g,h}(s, \hat{c}, a'))$.
14:         For all $(s, \hat{c}) \in \mathcal{S} \times \mathcal{C}$, update $V_{r,h}(s, \hat{c}) \leftarrow \langle Q_{r,h}(s, \hat{c}, \cdot), \pi_h^k(\cdot|s,\hat{c})\rangle$, $V_{g,h}(s, \hat{c}) \leftarrow \langle Q_{g,h}(s, \hat{c}, \cdot), \pi_h^k(\cdot|s,\hat{c})\rangle$.
15:     **end for**
16:     $\hat{\tau}_k = \arg\max_{\hat{c} \in \mathcal{C}}[V_{r,1}(s_1, \hat{c}) + \lambda_k(\hat{c} + V_{g,1}(s_1, \hat{c}))]$.
17:     $\lambda_{k+1} = \text{Proj}_{[0,\xi]}(\lambda_k + \eta(\mathcal{B} - (\hat{\tau}_k + V_{g,1}(s_1, \hat{\tau}_k))))$
18:     **for** step $h = 1$ to $H$ **do**
19:         Take action $a_h^k$ according to the greedy policy $\pi_h^k$ starting from initial augmented state $(s_1, \hat{\tau}_k)$.
20:         Insert $(s_h^k, a_h^k, s_{h+1}^k)$ into $\mathcal{D}_h$
21:     **end for**
22: **end for**

---

**Lemma 5.2.** *For any $\lambda \in [0, \xi]$, we have*

$$\sum_{k=1}^{K}(\lambda - \lambda_k)(\mathcal{B} - (\hat{\tau}_k + V_{g,1}^k(x, \hat{\tau}_k))) \leq \frac{\lambda^2}{2\eta} + \frac{\eta V_{g,max}^2 K}{2}.$$

**Step 3**: We use optimistic estimates to bound $\mathcal{T}_2$.

**Lemma 5.3.** *With probability at least $1 - \delta$,*

$$\sum_k(V_{r,1}^{\pi^*}(x, \tau_*) - V_{r,1}^k(x, \hat{\tau}_k))+$$
$$\lambda_k((\tau_* + V_{g,1}^{\pi^*}(x, \tau_*)) - (\hat{\tau}_k + V_{g,1}^k(x, \hat{\tau}_k))) \leq \epsilon_0 K \xi.$$

Proving the optimism is challenging due to the continuity of the initial budget $\tau$. In the augmented MDP for the unconstrained CVaR problem, (Wang et al., 2023) leverages the fact that the greedy policy is optimal for the reward value function. In the unconstrained setting, one can bound the term $|(P_h^k(s,a) - \mathbb{P}_h(s,a))^T V_{r,h}^{\pi^*}(\cdot, c - g_h)|$ by showing that value function class on the greedy policy has small covering, as the Bellman operator is a contraction under the max operator. However, in the constrained setting considered here, the optimal policy is not necessarily greedy with

respect to the composite state-action value function. As a result, this approach does not apply.

Instead, we bound $|(P_h^k(s,a) - \mathbb{P}_h(s,a))^T V_{j,h}^k(\cdot, \hat{c})|$ for $j \in \{g, r\}$ using the uniform concentration bound from (Jin et al., 2020), and a discretization of the budget variable at resolution $\epsilon_0$, denoted by $\mathcal{C}$ (see Lemma D.1). The size of $\mathcal{C}$ appears in the uniform concentration bound and is reflected in the bonus term in line 8 of Algorithm 1. Specifically, given the boundedness of both the reward and utility value functions, we apply a standard $\varepsilon$-covering number argument to show that with high probability, at episode $k$, the composite value function of the optimal policy is bounded above by the optimistic estimate, up to a discretization gap of $\epsilon_0 \lambda_k$.

**Step 4**: The upper bound for $\mathcal{T}_1$ is established by leveraging the uniform concentration bound, applying Azuma's inequality, and invoking the Elliptical Potential Lemma.

**Lemma 5.4.** *With probability at least $1 - \delta$,*

$$\sum_k(V_{r,1}^k(x, \hat{\tau}_k) - V_{r,1}^{\pi^k}(x, \hat{\tau}_k))$$
$$\leq 20HS\sqrt{HAK|\mathcal{C}|\log(K)\log(HSAK|\mathcal{C}|/\delta)}.$$

**Step 5**: We bound $\mathcal{T}_4$ using a similar argument to the one used for bounding $\mathcal{T}_1$.

**Lemma 5.5.** *With probability at least* $1 - \delta$,

$$\sum_k ((\hat{\tau}_k + V_{g,1}^k(x, \hat{\tau}_k)) - (\hat{\tau}_k + V_{g,1}^{\pi^k}(x, \hat{\tau}_k)))$$

$$\leq 14 V_{g,\max} S \sqrt{HAK|\mathcal{C}| \log(K) \log(HSA|\mathcal{C}|V_{g,\max}/\delta)}$$
$$+ 14 V_{g,\max} S \log(K) \sqrt{HAK|\mathcal{C}|} + K\epsilon_0 H e^{|\alpha|H}.$$

Note that the learned policy $\pi^k$ is Markovian with respect to the discretized augmented state space, as it depends on the accumulated discretized utility rather than the exact cumulative utility. Consequently, $\pi^k$ is not Markovian with respect to the augmented state space which lies in the continuous space. This discrepancy is captured by the additional term in the upper bound of Lemma 5.5.

**Regret Bound**: We derive the regret bound by combining the bounds for $\mathcal{T}_1$, $\mathcal{T}_2$, and $\mathcal{T}_3$, and by setting the parameters as $\lambda = 0$, $\eta = K^{-1/4}/V_{g,max}$, $\epsilon_0 = K^{-1/2}$, and $\xi = K^{1/4}$. More specifically, the regret associated with the augmented CMDP is bounded by $\tilde{\mathcal{O}}(V_{g,\max} K^{3/4} + \sqrt{H^4 S^2 A \log(1/\delta)} K^{3/4})$.

**Violation Bound**: To bound the violation, we combine the bounds for $\mathcal{T}_1$, $\mathcal{T}_2$, and $\mathcal{T}_3$ for a fixed $\lambda$, by setting $\lambda = K^{1/4}$, $\eta = K^{-1/4}/V_{g,max}$, $\epsilon_0 = K^{-1/2}$, and $\xi = K^{1/4}$. In particular, for any fixed $\lambda \in [0, \xi]$, we have

$$\sum_k (V_{r,1}^{\pi^*}(x, \tau_*) - V_{r,1}^{\pi^k}(x, \hat{\tau}_k)) + \lambda(\mathcal{B} - (\hat{\tau}_k + V_{g,1}^{\pi^k}(x, \hat{\tau}_k)))$$

$$\leq \tilde{\mathcal{O}}(V_{g,\max} \sqrt{H^3 S^2 A \log(1/\delta)} K)$$

Note that $\left| \sum_k (V_{r,1}^{\pi^*}(x, \tau_*) - V_{r,1}^{\pi^k}(x, \hat{\tau}_k)) \right|$ is trivially bounded by $HK$. Setting $\lambda = \xi = K^{1/4}$, the violation associated with the augmented CMDP is bounded by $\tilde{\mathcal{O}}(V_{g,\max} \sqrt{H^3 S^2 A \log(1/\delta)} K^{3/4})$.

**From Augmented CMDP to Original CMDP**: As noted earlier, the policy $\pi^k$, together with the initial budget $\hat{\tau}_k$, can be interpreted as a history-dependent policy in the original CMDP. This policy depends on the current state and the accumulated discretized utility. To highlight the dependence of $\pi^k$ on $\hat{\tau}_k$, we denote it by $\pi^k(\hat{\tau}_k)$. Consequently, the reward associated with the policy $\pi^k(\hat{\tau}_k)$ is given by $V_{r,1}^{\pi^k}(s_1, \hat{\tau}_k)$ in both the original and the augmented CMDP. However, the entropic risk measure corresponding to this

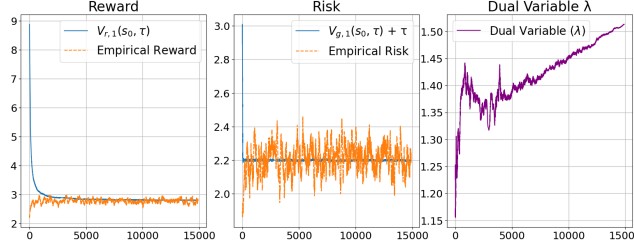

*Figure 1.* Evolution of the empirical reward $V_{r,1}^k(s_1, \hat{\tau}_k)$, the empirical risk measure for utility $\hat{\tau}_k + V_{g,1}^k(s_1, \hat{\tau}_k)$, and the dual variable $\lambda$ across iterations for $\alpha = -0.0001$. Here, $B = 2.2$. All values are averaged over the most recent 100 episodes.

policy, by Lemma 3.1, is given by

$$\max_\tau \left( \tau + \mathbb{E}_{\pi^k(\hat{\tau}_k)} \left[ u \left( \sum_{h=1}^H g_h(s_h, a_h) - \tau \,\bigg|\, s_1 \right) \right] \right)$$

$$\geq \hat{\tau}_k + \mathbb{E}_{\pi^k(\hat{\tau}_k)} \left[ u \left( \sum_{h=1}^H g_h(s_h, a_h) - \hat{\tau}_k \,\bigg|\, s_1 \right) \right]$$

$$= \hat{\tau}_k + V_{g,1}^{\pi^k}(s_1, \hat{\tau}_k).$$

Therefore, we have $\text{Regret}^{\text{aug}}(K) = \text{Regret}(K)$ and $\text{Violation}(K) \leq \text{Violation}^{\text{aug}}(K)$.

# 6. Simulation Results

We simulate our proposed approach on a $5 \times 5$ Grid-World with two actions: $\rightarrow$ and $\downarrow$. In all simulations, we use $K = 15,000$, $\xi = K^{-1/4}$, and $\eta = cK^{-1/4}/V_{g,\max}$, where the coefficient c is linearly scaled down from 100 to 1 across episodes to improve the convergence rate (i.e., a larger $\eta$ in earlier episodes). Further details of the simulation setup, including the reward, utility, and transition probabilities, are provided in Appendix I.

Figure 1 illustrates the behavior of the algorithm for $\alpha = -0.0001$ as a function of $k$. Initially, the policy violates the risk constraint and also underperforms in terms of reward. As the number of iterations increases, the empirical risk associated with the cumulative utility rises and eventually meets the threshold $B = 2.2$. The dual variable $\lambda$ increases rapidly in the early stages to penalize constraint violations, then grows more slowly once the policy becomes feasible. Additionally, we observe that the optimistic estimates $V_{r,1}^k(s_1, \hat{\tau}k)$ and $V_{g,1}^k(s_1, \hat{\tau}_k)$ are initially inflated due to exploration bonuses. As more data is gathered, these estimates converge to their empirical counterparts, and the gap between the estimated and empirical values gradually narrows.

Table 1 presents the empirical reward values $V_{r,1}^{\text{emp}}(s_1)$ and empirical risk values $V_{g,1}^{\text{emp}}(s_1)$ for the average policy induced by the final 20 episodes after $K = 15,000$ iterations,

*Table 1.* Empirical reward and risk from the average policy over the last 20 episodes after $K = 15{,}000$ iterations for various $(\alpha, B)$ pairs.

| | $(-10^{-2}, 2.2)$ | $(-10^{-4}, 2.2)$ | $(-10^{-2}, 2.6)$ | $(-10^{-4}, 2.6)$ | $(-10^{-2}, 2.9)$ | $(-10^{-4}, 2.9)$ |
|---|---|---|---|---|---|---|
| $V_{r,1}^{\mathrm{emp}}(s_0)$ | 1.95 | 2.80 | 1.90 | 2.24 | 1.92 | 1.88 |
| $V_{g,1}^{\mathrm{emp}}(s_0)$ | 2.76 | 2.20 | 2.80 | 2.57 | 2.78 | 2.80 |

across various choices of $\alpha$ and $B$. Decreasing $B$ enlarges the feasible set, enabling the algorithm to attain higher rewards while satisfying the constraint. Conversely, for fixed $B$, decreasing $\alpha$ shrinks the feasible set, resulting in lower achievable rewards. Figure 3 (Appendix I) shows that as $\alpha$ decreases how the policy selects less risky actions compromising the reward. When $|\alpha|$ and $B$ are large, the algorithm may fail to find a feasible policy, leading to constraint violations. Moreover, as $|\alpha|$ increases, the value of $V_{g,\max}$ also grows, which may impact both regret and constraint satisfaction. Thus, larger values of $|\alpha|$ may require more episodes ($K$) for a reliable performance evaluation.

## 7. Extension

As noted, this is the first work to establish theoretical bounds for online learning in a CMDP with a risk-sensitive constraint. Our analysis naturally extends to several related settings, some of which we outline below.

**Extension to other risk constraints**: In this paper, we consider a constraint requiring the entropic risk measure of the utility value function to exceed a specified threshold. Our analysis can be extended to other risk-sensitive constraints, such as Conditional Value at Risk (CVaR) with a hard threshold, by adopting a similar augmented formulation as in the unconstrained setting (Wang et al., 2024; Liang & Luo, 2024). A complete characterization of regret and constraint violation in these cases is left for future work.

**Extension to risk-sensitive objective**: In this paper, we focus on the objective of maximizing the expected cumulative reward. However, our analysis can be extended to settings where the goal is to maximize the entropic risk measure of the reward value function. This extension requires augmenting the state space with two budget variables: $\tau_r$ for the reward and $\tau_g$ for the utility. The optimization proceeds in two stages: first, for each $\tau_r$, we optimize over $\tau_g$ to evaluate the entropic risk of the utility; then, we optimize over $\tau_r$ to bound the entropic risk of the reward. A full characterization of this framework is left for future work.

**Extension to Multiple constraints**: Our framework can be extended to accommodate multiple risk-sensitive constraints. Specifically, we augment the state space with multiple budget variables $(\tau_1, \ldots, \tau_I)$, where $I$ denotes the number of constraints. For the resulting augmented state space, we first compute the value functions associated with each budget, and then jointly optimize over the initial budgets $(\tau_1, \ldots, \tau_I)$. However, this extension influences the regret

and violation bounds, which will scale with the number of constraints. A complete characterization of this extended framework is left for future work.

## 8. Conclusion and Future Work

In this work, we study the problem of maximizing cumulative reward subject to the constraint that the entropic risk measure of the utility function remains above a specified threshold. This formulation is particularly relevant in safety-critical applications. We consider an online learning setting in which an agent interacts with the environment over $K$ episodes, aiming to minimize both regret and constraint violation. The problem presents significant challenges due to the non-linearity of the entropic risk value, which renders standard primal-dual methods difficult to analyze. To overcome this, we reformulate the problem by augmenting the state space with a budget variable. By optimizing over the initial budget, we show that the value function in the augmented formulation exactly recovers the entropic risk measure, allowing us to recast the problem into a structure amenable to value-based methods.

Since strong duality may not hold in this setting, we introduce a truncated dual update. We prove that the resulting algorithm achieves a regret bound of $\tilde{\mathcal{O}}\big(V_{g,\max}K^{3/4} + \sqrt{H^4 S^2 A \log(1/\delta)}K^{3/4}\big)$ and a constraint violation bound of $\tilde{\mathcal{O}}\big(V_{g,\max}\sqrt{H^3 S^2 A \log(1/\delta)}K^{3/4}\big)$, with probability at least $1 - \delta$. To the best of our knowledge, this is the first result establishing theoretical bounds for the entropic risk-constrained MDP problem.

An important direction for future work is determining whether tighter violation bounds, particularly with improved dependence on $K$, can be achieved. Recent approaches (Jiang & Ye, 2024; Dalal et al., 2018) to improve the sample complexity bound for risk-neutral CMDP setting can be useful in reducing the dependency. Extending the analysis to settings with infinite state spaces remains an open challenge. Another promising avenue is to consider environments where utilities and rewards are drawn from unknown distributions, introducing additional complexity in both learning and generalization. Another important future direction is to consider a stricter violation bound, such as no cancellation violation bound as considered in risk-neutral CMDP setting (Ghosh et al., 2024). Empirical evaluations for larger state-space, and for a more practical setup have been left for the future.

## Impact Statement

This paper is theoretical in nature. The goal of this work is to advance the safe decision making using RL. We do not foresee any immediate or unique ethical concerns that require specific attention in this context.

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

## A. Limitations of Markov Policies under Entropic Risk Constraint

Here, we present a simple example that illustrates the limitations of Markov policies in a CMDP with an entropic risk measure. In particular, the optimal policy in this example does not lie within the class of Markov policies. This demonstrates that Markov policies can be suboptimal, underscoring the need to extend the policy search to history-dependent policies in our framework.

Consider a CMDP $(\mathcal{S}, \mathcal{A}, \mathbb{P}, H, r, g)$, where the state space is $\mathcal{S} = \{s_1, s_2, s_2', s_3\}$, the action space is $\mathcal{A} = \{a, b\}$, and the horizon is $H = 3$. The transition probabilities are depicted in Figure 2 and are independent of the actions taken at the states. The reward and utility functions are zero everywhere except at the following state-action pairs:

$$r(s_2, \cdot) = 1, \quad g(s_2', \cdot) = 1, \quad r(s_3, a) = 1, \quad g(s_3, b) = 1.$$

Notice that action $a$ maximizes the expected reward, while action $b$ maximizes the entropic risk measure. Let the lower bound on acceptable utility be $\mathcal{B} = 1$ in (1). Consider a history-dependent policy that selects action $b$ if and only if the state at $h = 1$ is $s_1$. It is easy to verify that this policy is feasible, and the resulting expected reward is 1.

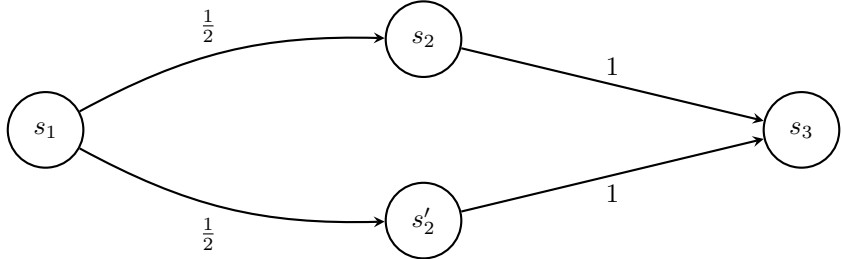

*Figure 2.* Transition diagram for the CMDP.

Next, we show that any Markov policy is suboptimal. Let $\boldsymbol{\pi} = (\pi_0, \pi_1, \pi_2)$ denote a Markov policy. Observe that the only relevant components are $\pi_2(a \,|\, s_2)$ and $\pi_2(b \,|\, s_2)$. Define $p = \pi_2(a \,|\, s_2)$. Then we have:

$$V_{r,1}^{\pi}(s_1) = \frac{1}{2} + p,$$
$$V_{g,1}^{\pi}(s_1) = \frac{1}{\alpha} \log \left( \frac{p}{2} + \frac{p}{2} e^{\alpha} + \frac{1-p}{2} e^{\alpha} + \frac{1-p}{2} e^{2\alpha} \right) < \frac{3}{2} - p.$$

where $\alpha < 0$ is the risk factor, and the last inequality follows by Jensen's inequality. Observe that for any $p < \frac{1}{2}$, the resulting policy is suboptimal since $V_{r,1}^{\pi}(s_1) < 1$. On the other hand, for any $p \geq \frac{1}{2}$, we have $V_{g,1}^{\pi}(s_1) < 1$, and thus the resulting policy is infeasible.

## B. Proof of Lemma 3.1

From the definition, $\text{OCE}(\pi) = \max_\tau \left( \tau + \mathbb{E} \left[ u \left( \sum_{h=1}^{H} g_h(s_h, a_h) - \tau \right) \right] \right)$, where $u(t) = \frac{1}{\alpha}(e^{\alpha t} - 1)$. Note that for $\alpha < 0$, the function $u(t)$ is strictly concave, hence, the unique maximizer can be found using the first order condition:

$$\tau_* = \frac{1}{\alpha} \log \left( \mathbb{E} \left[ \exp \left( \alpha \sum_{h=1}^{H} g_h(s_h, a_h) \right) \right] \right).$$

The result follows by substituting $\tau_*$ in the definition of $\text{OCE}(\pi)$.

## C. Proof of Lemma 5.2

Note that for $\lambda \in [0, \xi]$, we have

$$
\begin{aligned}
|\lambda_{k+1} - \lambda|^2 &= |\text{Proj}_{[0,\xi]}(\lambda_k + \eta(\mathcal{B} - (\hat{\tau}_k + V_{g,1}(s_1, \hat{\tau}_k)))) - \text{Proj}(\lambda)|^2 \\
&\leq |\lambda_k + \eta(\mathcal{B} - (\hat{\tau}_k + V_{g,1}(s_1, \hat{\tau}_k))) - \lambda|^2 \\
&\leq |\lambda - \lambda_k|^2 - 2\eta(\mathcal{B} - (\hat{\tau}_k + V_{g,1}(s_1, \hat{\tau}_k)))(\lambda - \lambda_k) + \eta^2 V_{g,max}^2
\end{aligned}
$$

where the first inequality follows from the non-expansiveness of the projection operator, and the second from the fact that $|\mathcal{B} - (\hat{\tau}_k + V_{g,1}(s_1, \hat{\tau}_k))| \leq V_{g,\max}$. Summing over $k$ then yields

$$
0 \leq |\lambda_{K+1} - \lambda|^2 \leq |\lambda_1 - \lambda|^2 - \sum_k 2\eta(\mathcal{B} - (\hat{\tau}_k + V_{g,1}(s_1, \hat{\tau}_k)))(\lambda - \lambda_k) + K\eta^2 V_{g,max}^2
$$

Since $\lambda_1 = 0$, it follows that

$$
\sum_{k=1}^{K}(\lambda - \lambda_k)(\mathcal{B} - (\hat{\tau}_k + V_{g,1}^k(x, \hat{\tau}_k))) \leq \frac{\lambda^2}{2\eta} + \frac{\eta V_{g,max}^2 K}{2}
$$

## D. Proof of Lemma 5.3

Recall that $\pi^* = \{\pi_h^*\}_{h=1}^H$ denotes the optimal feasible policy that solves the optimization problem (1). By assumption, there exists a corresponding Markovian policy in the augmented state space that solves the optimization problem (3). With slight abuse of notation, we also denote this Markovian policy by $\pi^* = \{\pi_h^*\}_{h=1}^H$, where each $\pi_h^* : \mathcal{S} \times [-H, H] \to \mathcal{A}$. As noted earlier, the budget variable $c_h$ in the augmented MDP at time $h \geq 1$ lies in the interval $[-H, H]$.

To improve computational efficiency and address the challenge of optimizing the initial budget $\hat{\tau}_k$, we discretize both the initial budget and the utility values received at each state. Consequently, the learned policy $\pi^k = \{\pi_h^k\}_{h=1}^H$ depends on the accumulated discretized utility rather than the exact cumulative utility. As a result, $\pi^k$ is not Markovian with respect to the augmented state space, but instead is Markovian with respect to the discretized augmented state space, where all utility values are replaced by their discretized counterparts.

Specifically, the budget variable in the discretized augmented state space is restricted to a finite set $\mathcal{C}$, which is an $\epsilon_0$-resolution discretization of the interval $[-H, H]$. Thus, the cardinality of $\mathcal{C}$ is given by

$$
|\mathcal{C}| = \left\lceil \frac{2H}{\epsilon_0} \right\rceil.
$$

Finer discretization improves the approximation accuracy at the cost of increased computational complexity. The discretization operator $\phi : [-H, H] \to \mathcal{C}$ projects a real-valued budget onto the nearest larger discretized value:

$$
\phi(c) = \arg\min_{\hat{c} \in \mathcal{C}, \, \hat{c} \geq c} |\hat{c} - c|.
$$

For notational consistency, all hatted symbols denote discretized values. In particular, we use $\hat{g}_h(s, a)$ as a shorthand for $\phi(g_h(s, a))$.

**Lemma D.1.** *For all* $(s, a) \in \mathcal{S} \times \mathcal{A}$, $\hat{c} \in \mathcal{C}$, $h \leq H$, $k \leq K$, *and* $j \in \{r, g\}$

$$
\left| \left( P_h^k(\cdot|s, a) - P_h(\cdot|s, a) \right)^T V_{j,h}^k(\cdot, \hat{c}) \right| \leq \text{Bon}_{j,h}^k(s, a)
$$

*with probability at least* $1 - \delta$, *where* $\text{Bon}_{j,h}^k(s, a) = 6V_{j,\max}\sqrt{S|\mathcal{C}| \log\left(HSAK|\mathcal{C}|V_{j,\max}/\delta\right)/N_h^k(s, a)}$, $V_{r,\max} = H$, *and* $V_{g,\max} = \dfrac{\exp(|\alpha|H) - 1}{|\alpha|}$.

**Lemma D.2.** *For all $(s,a) \in \mathcal{S} \times \mathcal{A}$, $c \in [-H, H]$, $\hat{c} \in \mathcal{C}$, $h \leq H$, and $k \leq K$, and $j \in \{r, g\}$, the following holds with probability at least $1 - \delta$:*

$$Q_{j,h}^{\pi^*}(s, c, a) - Q_{j,h}^k(s, \hat{c}, a) \leq (P_h(\cdot \mid s, a))^T \left( V_{j,h+1}^{\pi^*}(\cdot, c - g_h(s, a)) - V_{j,h+1}^k(\cdot, \hat{c} - \hat{g}_h(s, a)) \right).$$

*Proof.*

$$\begin{aligned}
Q_{j,h}^{\pi^*}(s, c, a) - Q_{j,h}^k(s, \hat{c}, a) &= (P_h(\cdot \mid s, a))^T V_{j,h+1}^{\pi^*}(\cdot, c - g_h(s, a)) \\
&\quad - \mathrm{Bon}_{j,h}^k(s, a) - \left( P_h^k(\cdot \mid s, a) \right)^T V_{j,h+1}^k(\cdot, \hat{c} - \hat{g}_h(s, a)) \\
&= (P_h(\cdot \mid s, a))^T \left( V_{j,h+1}^{\pi^*}(\cdot, c - g_h(s, a)) - V_{j,h+1}^k(\cdot, \hat{c} - \hat{g}_h(s, a)) \right) \\
&\quad - \mathrm{Bon}_{j,h}^k(s, a) - \left( P_h^k(\cdot \mid s, a) - P_h(\cdot \mid s, a) \right)^T V_{j,h+1}^k(\cdot, \hat{c} - \hat{g}_h(s, a)) \\
&\leq (P_h(\cdot \mid s, a))^T \left( V_{j,h+1}^{\pi^*}(\cdot, c - g_h(s, a)) - V_{j,h+1}^k(\cdot, \hat{c} - \hat{g}_h(s, a)) \right) \\
&\quad - \mathrm{Bon}_{j,h}^k(s, a) + \mathrm{Bon}_{j,h}^k(s, a),
\end{aligned}$$

where the last inequality follows by Lemma D.1. $\square$

**Lemma D.3.** *For all $(s,a) \in \mathcal{S} \times \mathcal{A}$, $c \in [-H, H]$, $\hat{c} \in \mathcal{C}$ with $\hat{c} \leq c$, $h \leq H$, and $k \leq K$, the following holds with probability at least $1 - \delta$:*

$$Q_{r,h}^{\pi^*}(s, c, a) + \lambda_k Q_{g,h}^{\pi^*}(s, c, a) - Q_{r,h}^k(s, \hat{c}, a) - \lambda_k Q_{g,h}^k(s, \hat{c}, a) \leq 0$$

*Proof.* We prove the claim by induction. The base case holds trivially for $h = H + 1$, since $Q_{r,H+1}^{\pi^*}(s, c, a) = Q_{r,H+1}^k(s, \hat{c}, a) = 0$ and $Q_{g,H+1}^{\pi^*}(s, c, a) = u(-c) \leq u(-\hat{c}) = Q_{g,H+1}^k(s, \hat{c}, a)$, where the inequality follows from the monotonicity of $u(\cdot)$. For the induction step, assuming the statement holds for horizon $h + 1$, we will verify it for horizon $h$:

$$\begin{aligned}
Q_{r,h}^{\pi^*}(s, c, a) &+ \lambda_k Q_{g,h}^{\pi^*}(s, c, a) - Q_{r,h}^k(s, \hat{c}, a) - \lambda_k Q_{g,h}^k(s, \hat{c}, a) \\
&\leq (P_h(\cdot \mid s, a))^T \left( V_{r,h+1}^{\pi^*}(\cdot, c - g_h(s, a)) - V_{r,h+1}^k(\cdot, \hat{c} - \hat{g}_h(s, a)) \right) \\
&\quad + \lambda_k (P_h(\cdot \mid s, a))^T \left( V_{g,h+1}^{\pi^*}(\cdot, c - g_h(s, a)) - V_{g,h+1}^k(\cdot, \hat{c} - \hat{g}_h(s, a)) \right) \\
&\leq 0
\end{aligned}$$

where the first inequality follows from Lemma D.2, and the last inequality follows from the induction hypothesis, the assumption that $\hat{c} \leq c$, and the inequality $\hat{g}_h(s, a) \geq g_h(s, a)$. Notice that by the definition of $\pi^k$,

$$\begin{aligned}
V_{r,h+1}^{\pi^*}(s, c) &+ \lambda_k V_{g,h+1}^{\pi^*}(s, c) - (V_{r,h+1}^k(s, \hat{c}) + \lambda_k V_{g,h+1}^k(s, \hat{c})) \\
&\leq \sum_{a \in \mathcal{A}} \pi^*(a|s, c) \left( Q_{r,h+1}^{\pi^*}(s, c, a) + \lambda_k Q_{g,h+1}^{\pi^*}(s, c, a) - Q_{r,h+1}^k(s, \hat{c}, a) - \lambda_k Q_{g,h+1}^k(s, \hat{c}, a) \right).
\end{aligned}$$

$\square$

As an immediate corollary of Lemma D.3, the following inequality holds with probability at least $1 - \delta$:

$$\begin{aligned}
V_{r,1}^{\pi^*}(s_1, \tau_*) &+ \lambda_k \left( \tau_* + V_{g,1}^{\pi^*}(s_1, \tau_*) \right) - V_{r,1}^k(s_1, \hat{\tau}_k) - \lambda_k \left( \hat{\tau}_k + V_{g,1}^k(s_1, \hat{\tau}_k) \right) \\
&\leq V_{r,1}^{\pi^*}(s_1, \tau_*) + \lambda_k \left( \tau_* + V_{g,1}^{\pi^*}(s_1, \tau_*) \right) - V_{r,1}^k(s_1, \phi(\tau_*) - \epsilon_0) - \lambda_k \left( \phi(\tau_*) - \epsilon_0 + V_{g,1}^k(s_1, \phi(\tau_*) - \epsilon_0) \right) \\
&\leq \lambda_k \epsilon_0
\end{aligned}$$

where we use the fact that the initial budget at episode $k$ is given by $\hat{\tau}_k = \arg\max_{\hat{c} \in \mathcal{C}} \left[ V_{r,1}^k(s_1, \hat{c}) + \lambda_k \left( \hat{c} + V_{g,1}^k(s_1, \hat{c}) \right) \right]$, and the inequality $\phi(\tau_*) - \epsilon_0 < \tau_* \leq \phi(\tau_*)$. Finally, observe that $\lambda_k \leq \xi$.

# E. Proof of Lemma 5.4 and Lemma 5.5

As noted earlier, the policy $\pi^k = \{\pi_h^k\}$ is Markovian with respect to the discretized augmented state space, but history-dependent with respect to the augmented state space. However, note that $V_{g,H+1}^{\pi^k}$ depends on the cumulative utility, while $V_{g,H+1}^k$ depends on the cumulative discretized utility. More specifically,

$$\pi_h^k(\cdot \mid s_h, c_h, \{s_{h'}, a_{h'}, c_{h'}\}_{h'=1}^{h-1}) = \pi_h^k\left(\cdot \mid s_h, \sum_{h'=1}^{h} \phi(g_{h'}(s_{h'}, a_{h'}))\right),$$

$$V_{g,H+1}^k(\{s_{h'}, a_{h'}, c_{h'}\}_{h'=1}^H) = u\left(-\phi(c_1) + \sum_{h'=1}^{H} \phi(g_{h'}(s_{h'}, a_{h'}))\right),$$

$$V_{g,H+1}^{\pi^k}(\{s_{h'}, a_{h'}, c_{h'}\}_{h'=1}^H) = u\left(-c_1 + \sum_{h'=1}^{H} g_{h'}(s_{h'}, a_{h'})\right).$$

In particular, the value of $V_{g,h}^{\pi^k}$ depends on both the cumulative discretized utility (since the policy $\pi^k$ depends on it) and the cumulative utility (since $V_{g,H+1}^{\pi^k}$ depends on it). For notational simplicity, we retain only the cumulative discretized utility in the notation and explicitly acknowledge the dependence on the cumulative utility when referring to $V_{g,H+1}^{\pi^k}$. In what follows, we adopt the convention of reserving the " ˆ " symbol exclusively for discretized values.

**Lemma E.1.** *The following inequality hold with probability at least $1 - \delta$:*

$$V_{r,1}^k(s, \hat{\tau}_k) - V_{r,1}^{\pi^k}(s, \hat{\tau}_k) \leq 2 \sum_{h=1}^{H} \mathbb{E}_{\pi^k}[\text{Bon}_{r,h}^k(s_h, a_h) \mid s_1 = s, \hat{c}_1 = \hat{\tau}_k, \mathcal{H}_k]$$

$$V_{g,1}^k(s, \hat{\tau}_k) - V_{g,1}^{\pi^k}(s, \hat{\tau}_k) \leq 2 \sum_{h=1}^{H} \mathbb{E}_{\pi^k}[\text{Bon}_{g,h}^k(s_h, a_h) \mid s_1 = s, \hat{c}_1 = \hat{\tau}_k, \mathcal{H}_k] + \epsilon_0 H e^{|\alpha|H}$$

*Proof.* For all $s \in \mathcal{S}, \hat{c} \in \mathcal{C}, h \leq H, k \leq K$, and $j \in \{r, g\}$ the following holds with probability at least $1 - \delta$:

$$V_{j,h}^k(s, \hat{c}) - V_{j,h}^{\pi^k}(s, \hat{c}) = \sum_{a \in \mathcal{A}} \pi^k(a|s, \hat{c}) \left(Q_{j,h}^k(s, \hat{c}, a) - Q_{j,h}^{\pi^k}(s, \hat{c}, a)\right)$$

$$= \sum_{a \in \mathcal{A}} \pi^k(a|s, \hat{c}) \left(\text{Bon}_{j,h}^k(s, a) + \left(P_h^k(\cdot|s, a)\right)^T V_{j,h+1}^k(\cdot, \hat{c} - \hat{g}_h(s, a)) - Q_{j,h}^{\pi^k}(s, \hat{c}, a)\right)$$

$$= \mathbb{E}_{a \sim \pi^k(\cdot|s, \hat{c})} \left[\text{Bon}_{j,h}^k(s, a) + \left(P_h^k(\cdot|s, a) - P_h(\cdot|s, a)\right)^T V_{j,h+1}^k(\cdot, \hat{c} - \hat{g}_h(s, a))\right]$$

$$+ \sum_{a \in \mathcal{A}} \pi^k(a|s, \hat{c}) \left(P_h(\cdot|s, a)\right)^T \left(V_{j,h+1}^k(\cdot, \hat{c} - \hat{g}_h(s, a)) - V_{j,h+1}^{\pi^k}(\cdot, \hat{c} - \hat{g}_h(s, a))\right)$$

$$\leq 2\mathbb{E}_{a \sim \pi^k(\cdot|s, \hat{c})}\left[\text{Bon}_{j,h}^k(s, a)\right] + \mathbb{E}_{\pi^k}\left[V_{j,h+1}^k(s_{h+1}, \hat{c}_{h+1}) - V_{j,h+1}^{\pi^k}(s_{h+1}, \hat{c}_{h+1})\big|s_h = s, \hat{c}_h = \hat{c}\right],$$

where we used the fact that $\pi^k$ is Markovian with respect to the discretized augmented state space. Hence, we have

$$V_{r,1}^k(s, \hat{\tau}_k) - V_{r,1}^{\pi^k}(s, \hat{\tau}_k) \leq 2 \sum_{h=1}^{H} \mathbb{E}_{\pi^k}[\text{Bon}_{r,h}^k(s_h, a_h) \mid s_1 = s, c_1 = \hat{\tau}_k, \mathcal{H}_k],$$

$$V_{g,1}^k(s, \hat{\tau}_k) - V_{g,1}^{\pi^k}(s, \hat{\tau}_k) \leq 2 \sum_{h=1}^{H} \mathbb{E}_{\pi^k}[\text{Bon}_{g,h}^k(s_h, a_h) \mid s_1 = s, c_1 = \hat{\tau}_k, \mathcal{H}_k]$$

$$+ \mathbb{E}_{\pi^k}\left[u\left(-\hat{\tau}_k + \sum_{h'=1}^{H} \hat{g}_{h'}(s_{h'}, a_{h'})\right) - u\left(-\hat{\tau}_k + \sum_{h'=1}^{H} g_{h'}(s_{h'}, a_{h'})\right)\bigg|s_1 = s, c_1 = \hat{\tau}_k, \mathcal{H}_k\right],$$

where $\mathcal{H}_k$ is the trajectory from episodes $1, 2, \cdots, k - 1$. The result follows from the fact that $u(t)$, for $t \in [-H, H]$, is a Lipschitz function with Lipschitz constant $e^{|\alpha|H}$. $\square$

Combining Lemma E.1 with Elliptical Potential Lemma, we have

$$\sum_{k=1}^{K} V_{r,1}^{k}(s,\hat{\tau}_k) - V_{r,1}^{\pi^k}(s,\hat{\tau}_k) \leq 2\sum_{k=1}^{K}\sum_{h=1}^{H} \mathbb{E}_{\pi^k}[\mathrm{Bon}_{r,h}^{k}(s_h,a_h) \mid s_1 = s, \hat{c}_1 = \hat{\tau}_k] + 2H\sqrt{HK\log(1/\delta)}$$

$$= 2\sum_{k=1}^{K}\sum_{h=1}^{H} \mathbb{E}_{\pi^k}\left[9H\sqrt{\frac{S|\mathcal{C}|\log\left(HSAK|\mathcal{C}|/\delta\right)}{N_h(s_h^k,a_h^k)}} \,\middle|\, s_1 = s, \hat{c}_1 = \hat{\tau}_k\right] + 2H\sqrt{HK\log(1/\delta)}$$

$$\leq 20HS\sqrt{HAK|\mathcal{C}|\log(K)\log\left(HSAK|\mathcal{C}|/\delta\right)}$$

Using a similar argument,

$$\sum_{k=1}^{K} V_{g,1}^{k}(s,\hat{\tau}_k) - V_{g,1}^{\pi^k}(s,\hat{\tau}_k) \leq 2\sum_{k=1}^{K}\sum_{h=1}^{H} \mathbb{E}_{\pi^k}[\mathrm{Bon}_{g,h}^{k}(s_h,a_h) \mid s_1 = s, \hat{c}_1 = \hat{\tau}_k] + K\epsilon_0 He^{|\alpha|H} + 2V_{g,\max}\sqrt{HK\log(1/\delta)}$$

$$\leq 14V_{g,\max}S\sqrt{HAK|\mathcal{C}|\log(K)\log\left(HSAK|\mathcal{C}|V_{g,\max}/\delta\right)} + K\epsilon_0 He^{|\alpha|H}$$

# F. Bounding the Regret and Violation

**Bounding the Regret**: We bound the regret by combining Lemmas 5.2, 5.3, and 5.4, and by setting $\lambda = 0$, $\eta = K^{-1/4}/V_{g,max}$, $\epsilon_0 = K^{-1/2}$, and $\xi = K^{1/4}$:

$$\mathrm{Regret}^{\mathrm{aug}}(K) \leq \eta V_{g,\max}^2 K + \epsilon_0 K^{5/4} + 20HS\sqrt{HAK|\mathcal{C}|\log(K)\log\left(HSAK|\mathcal{C}|/\delta\right)}$$

$$= \tilde{\mathcal{O}}\left(\left(\frac{e^{|\alpha|H}-1}{|\alpha|H}\right)HK^{3/4} + \sqrt{H^4 S^2 A\log(1/\delta)}K^{3/4}\right).$$

**Bounding the Violation**: We bound the violation by combining Lemmas 5.2, 5.3, 5.4, and 5.5 and by setting $\lambda = 0$, $\eta = K^{-1/4}/V_{g,max}$, $\epsilon_0 = K^{-1/2}$, and $\xi = K^{1/4}$. Note that for a fixed $\lambda \in [0,\xi]$, we have

$$\sum_k \left(V_{r,1}^{\pi^*}(x,\tau_*) - V_{r,1}^{\pi^k}(x,\hat{\tau}_k)) + \lambda(\mathcal{B} - (\hat{\tau}_k + V_{g,1}^{\pi^k}(x,\hat{\tau}_k))\right)$$

$$\leq \eta V_{g,\max}^2 K + \frac{\xi^2}{2\eta} + \epsilon_0 K\xi + 20HS\sqrt{HAK|\mathcal{C}|\log(K)\log\left(HSAK|\mathcal{C}|/\delta\right)}$$

$$\quad + 14\xi V_{g,\max}S\sqrt{HAK|\mathcal{C}|\log(K)\log\left(HSAK|\mathcal{C}|V_{g,\max}/\delta\right)} + K\epsilon_0 He^{|\alpha|H}$$

$$\leq V_{g,max}K^{3/4} + \frac{V_{g,max}}{2}K^{3/4} + K^{3/4} + 60K^{3/4}\sqrt{S^2 H^4 A\log(K)\log\left(HSAK/\delta\right)}$$

$$\quad + 42V_{g,\max}K\sqrt{S^2 H^3 A\log(K)\log\left(HSAK/\delta\right)} + K^{1/2}He^{|\alpha|H}$$

$$\leq \tilde{\mathcal{O}}\left(V_{g,\max}K\sqrt{S^2 H^3 A\log\left(1/\delta\right)}\right).$$

Trivially bounding $\sum_k (V_{r,1}^{\pi_k}(x,\hat{\tau}_k) - V_{r,1}^{\pi^*}(x,\tau^*)) \leq HK$, we have

$$\lambda\sum_K \left(\mathcal{B} - (\hat{\tau}_k + V_{g,1}^{\pi^k}(x,\hat{\tau}_k))\right) \leq HK + \tilde{\mathcal{O}}\left(V_{g,\max}K\sqrt{S^2 H^3 A\log\left(1/\delta\right)}\right)$$

Setting $\lambda = \xi = K^{1/4}$ and dividing both sides by $K^{1/4}$, we have

$$\sum_k \left(\mathcal{B} - (\hat{\tau}_k + V_{g,1}^{\pi^k}(x,\hat{\tau}_k))\right) \leq \tilde{O}\left(\left(\frac{e^{|\alpha|H}-1}{|\alpha|H}\right)K^{3/4}\sqrt{S^2 H^5 A\log\left(1/\delta\right)}\right)$$

## G. Proof of Lemma D.1

Consider the value function classes $\mathcal{V}_j = \{V(\cdot, \cdot) \mid V : \mathcal{S} \times \mathcal{C} \to [0, V_{j,\max}]\}$ for $j \in \{r, g\}$, where $V_{r,\max} = H$ and $V_{g,\max} = \dfrac{\exp(|\alpha|H) - 1}{|\alpha|}$. The $\varepsilon$-covering numbers of $\mathcal{V}_j$ under the $\ell_\infty$ norm is bounded by

$$\mathcal{N}(\mathcal{V}_j, \varepsilon, \|\cdot\|_\infty) \leq \left(1 + \frac{V_{j,\max}}{\varepsilon}\right)^{S|\mathcal{C}|},$$

where $\mathcal{S}$ is the state space, $S = |\mathcal{S}|$, and $\mathcal{C}$ is the discretized budget space. We provide a uniform concentration bound.

Fix $j \in \{r, g\}$, $(s, a) \in \mathcal{S} \times \mathcal{A}$, $\hat{c} \in \mathcal{C}$, $h \leq H$, and $k \leq K$. Pick a function $V \in \mathcal{V}_j$. Applying Azuma-Hoeffding's inequality, we obtain

$$\mathbb{P}\left(\left|\left(P_h^k(\cdot|s, a) - P_h(\cdot|s, a)\right)^T V(\cdot, \hat{c})\right| > \epsilon\right) \leq 2 \exp\left(-\frac{N_h^k(s, a)\epsilon^2}{2V_{j,\max}^2}\right),$$

where $N_h^k(s, a)$ denotes the number of visits to $(s, a)$ at time $h$ up to episode $k$. Hence, with probability at least $1 - \delta$,

$$\left|\left(P_h^k(\cdot|s, a) - P_h(\cdot|s, a)\right)^T V(\cdot, \hat{c})\right| < 2V_{j,\max}\sqrt{\frac{\log(1/\delta)}{N_h^k(s, a)}}.$$

Let $\mathcal{V}_\varepsilon \subset \mathcal{V}_j$ denote an $\varepsilon$-covering set. Applying a union bound together with a standard covering argument, with probability at least $1 - \delta/2$, for all $(s, a) \in \mathcal{S} \times \mathcal{A}$, $\hat{c} \in \mathcal{C}$, $h \leq H$, $k \leq K$, and $V \in \mathcal{V}_j$, we have

$$\begin{aligned}
\left|\left(P_h^k(\cdot|s, a) - P_h(\cdot|s, a)\right)^T V(\cdot, \hat{c})\right| &\leq 2\varepsilon + 2V_{j,\max}\sqrt{\frac{\log\left(2HSAK|\mathcal{C}|\mathcal{N}(\mathcal{V}_j, \varepsilon, \|\cdot\|_\infty)/\delta\right)}{N_h^k(s, a)}} \\
&\leq 2\varepsilon + 2V_{j,\max}\sqrt{\frac{S|\mathcal{C}|\log\left(2HSAK|\mathcal{C}|(1 + V_{j,\max}/\varepsilon)/\delta\right)}{N_h^k(s, a)}} \\
&\leq 6V_{j,\max}\sqrt{\frac{S|\mathcal{C}|\log\left(HSAK|\mathcal{C}|V_{j,\max}/\delta\right)}{N_h^k(s, a)}},
\end{aligned}$$

where the last inequality uses $\varepsilon = 1/K$ and the fact that $N_h^k(s, a) \leq K$. The final result follows by applying a union bound over $\mathcal{V}_g$ and $\mathcal{V}_r$. See (Agarwal et al., 2022)[Lemma 7.2] for a similar argument.

## H. Auxiliary Lemmas

We adopt the following results from (Wang et al., 2023).

**Lemma H.1** (Azuma). *Let $\{X_i\}_{i \in [N]}$ be a sequence of random variables supported on $[0, 1]$, adapted to the filtration $\{\mathcal{F}_i\}_{i \in [N]}$. For any $\delta \in (0, 1)$, we have with probability at least $1 - \delta$,*

$$\sum_{t=1}^N E[X_t | \mathcal{F}_{t-1}] = \sum_{t=1}^N X_t + \sqrt{N \log(2/\delta)} \tag{6}$$

**Lemma H.2** (Elliptical Potential). *For any sequence of states and actions $\{s_{h,k}, a_{h,k}\}_{h \in [H], k \in [K]}$, we have*

$$\sum_{k=1}^K \sum_{h=1}^H \frac{1}{N_k(s_{h,k}, a_{h,k})} \leq SA \log(K),$$

$$\sum_{k=1}^K \sum_{h=1}^H \frac{1}{\sqrt{N_k(s_{h,k}, a_{h,k})}} \leq \sqrt{HSAK \log(K)}.$$

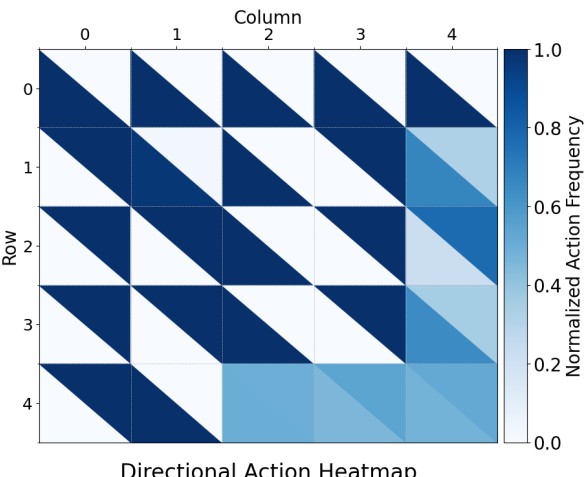 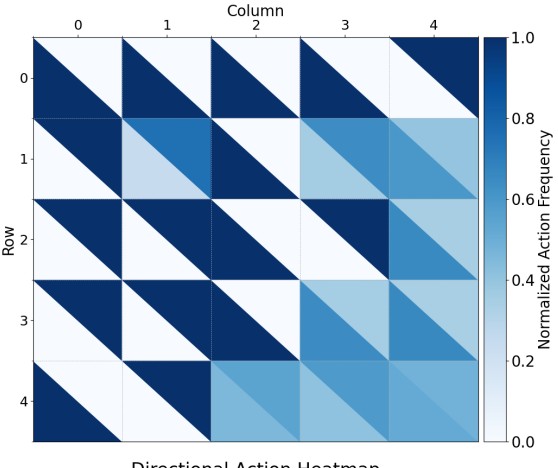

*Figure 3.* Frequency of actions for the average policy over the last 20 episodes after 15,000 iterations. Left: $\alpha = -0.01$, Right: $\alpha = -0.0001$, with $B = 2.2$.

## I. Simulation Environment

The reward function is given in Table 2, utility values in Table 3, and transition probabilities in Table 4. Both rewards and utilities depend on the state. The transition probability $p$ specifies the chance of successfully moving in the intended direction (e.g., choosing the $\rightarrow$ action attempts to move the agent to the right neighboring cell with probability $p$). If the intended move is invalid (e.g., outside the grid), an alternative action is executed. Consequently, in the fourth row and column, the choice of action becomes inconsequential. The total horizon is $H = 9$, and a state at position $(i, j)$ is only reachable at time step $h = i + j + 1$.

The environment presents several interesting features. For instance, the state at position $(1, 2)$ offers a high reward. However, it is surrounded by states with transition probabilities close to $0.5$ and low utility, making it risky. While it may appear attractive to visit $(1, 2)$, the agent might instead end up in one of the neighboring low-utility states. Consequently, as $\alpha$ becomes more negative, the agent may learn to avoid this region in order to mitigate risk which may lead to a smaller cumulative reward. The environment is deterministic across the terminal rows and columns. The code is available at: https://github.com/mmoharami/Risk-Sensitive-CMDP.

*Table 2.* Reward matrix $r(i, j)$ for state $(i, j)$

| Row \ Col | 0 | 1 | 2 | 3 | 4 |
|---|---|---|---|---|---|
| 0 | 0.0 | 0.1 | 0.2 | 0.2 | 0.1 |
| 1 | 0.5 | 0.1 | 1.5 | 0.5 | 0.3 |
| 2 | 0.1 | 0.1 | 0.4 | 0.3 | 0.2 |
| 3 | 0.1 | 0.1 | 0.3 | 0.1 | 0.6 |
| 4 | 0.1 | 0.2 | 0.3 | 0.1 | 0.0 |

For a faster convergence, we use the bonus terms $Bon^k_{r,h}(s, a) = 0.5H \log(K)/N^k_h(s, a)$, and $Bon^k_{g,h}(s, a) = 0.005V_{g,max} \log(K)/N^k_h(s, a)$ instead of the values for which we obtain the regret and the violation bound across all the values of $\alpha$. We use the discretized budget space with precision $K^{-1/2}$, and $\delta = 0.05$. The initial policy $\pi_0$ is uniform across the two actions for every augmented state.

Figure 3 illustrates the frequency of selecting each action ($\rightarrow$ or $\downarrow$) at every state for $\alpha = -0.01$ and $\alpha = -0.0001$, respectively, with $B = 2.2$. The displayed policies are the average policies over the last 20 episodes after running the algorithm for 15,000 iterations.

*Table 3.* Utility matrix $u(i,j)$ for state $(i,j)$.

| Row \ Col | 0 | 1 | 2 | 3 | 4 |
|:---:|:---:|:---:|:---:|:---:|:---:|
| 0 | 0.1 | 0.1 | 0.2 | 0.1 | 0.1 |
| 1 | 0.4 | 0.2 | 0.1 | 0.0 | 0.0 |
| 2 | 0.3 | 0.4 | 1.0 | 0.0 | 0.1 |
| 3 | 0.2 | 0.5 | 0.4 | 0.2 | 0.1 |
| 4 | 0.1 | 0.1 | 0.4 | 0.2 | 0.0 |

*Table 4.* Probability matrix $p(i,j)$ representing the likelihood that the action taken in state $(i,j)$ will occur.

| Row \ Col | 0 | 1 | 2 | 3 | 4 |
|:---:|:---:|:---:|:---:|:---:|:---:|
| 0 | 0.9 | 0.9 | 0.7 | 0.5 | 1.0 |
| 1 | 0.9 | 0.9 | 0.5 | 0.5 | 1.0 |
| 2 | 0.7 | 0.9 | 0.9 | 0.6 | 1.0 |
| 3 | 0.9 | 0.8 | 0.8 | 0.5 | 1.0 |
| 4 | 1.0 | 1.0 | 1.0 | 1.0 | 1.0 |

Observe that under the more risk-averse setting $\alpha = -0.01$, the algorithm tends to avoid the risky state $(1,2)$ to satisfy the constraint. In contrast, in the more risk-neutral scenario $\alpha = -0.0001$, the algorithm chooses to visit state $(1,2)$ more frequently by taking appropriate actions, thereby increasing the reward while still adhering to the constraint.

