# OpenReview forum: "Online Learning in Risk Sensitive constrained MDP"
_ICML.cc/2025/Conference — ICML 2025 poster_

### Official Review · Reviewer_DwFN · 2025-03-13

**Overall Recommendation:** 3

**Summary:**

The paper develops the first sublinear regret bound for the risk sensitive CMDP problems. In contrast to the classical CMDP, an additional risk measure is put on the left hand side of the constraint value, and it is required that the final value is greater than a threshold. The paper develops the clever idea of introducing an augmented variable to denote the budget constraint and the optimizing over the minimax formulation of the CMDP, with an additional augmented state. Finally, the sublinear regret has been developed for the objective value, as well as the constraint.

**Claims And Evidence:**

The claims look good to me.

**Essential References Not Discussed:**

The algorithm developed in the paper is based on the Lagrangian formulation, where the primal-dual algorithmic framework is widely developed to solve it. However, primal-based algorithms have also been developed to solve the CMDP problems. For example, the primal-based algorithm has been developed in [1], [2], [3], [4], and [5]. In particular, the paper [6] develops resolving primal LP methods to solve the CMDP and achieves the first instance-dependent $\tilde{O}(1/\epsilon)$ sample complexity. I think these papers are worth mentioning in the literature.

References:

[1]. Yongshuai Liu, Jiaxin Ding, and Xin Liu. Ipo: Interior-point policy optimization under constraints. In Proceedings of the AAAI conference on artificial intelligence, volume 34, pages 4940–4947, 2020.

[2]. Yinlam Chow, Mohammad Ghavamzadeh, Lucas Janson, and Marco Pavone. Risk-constrained reinforcement learning with percentile risk criteria. Journal of Machine Learning Research, 18(167):1–51, 2018.

[3]. Yinlam Chow, Ofir Nachum, Aleksandra Faust, Edgar Duenez-Guzman, and Mohammad Ghavamzadeh. Lyapunov-based safe policy optimization for continuous control. arXiv preprint arXiv:1901.10031, 2019.

[4]. Gal Dalal, Krishnamurthy Dvijotham, Matej Vecerik, Todd Hester, Cosmin Paduraru, and Yuval Tassa. Safe exploration in continuous action spaces. arXiv preprint arXiv:1801.08757, 2018.

[5]. Tengyu Xu, Yingbin Liang, and Guanghui Lan. Crpo: A new approach for safe reinforcement learning with convergence guarantee. In International Conference on Machine Learning, pages 11480–11491. PMLR, 2021.

[6]. Jiang, Jiashuo, and Yinyu Ye. "Achieving $\tilde{O}(1/\epsilon)$ Sample Complexity for Constrained Markov Decision Process." NeurIPS, 2024.

**Experimental Designs Or Analyses:**

There is no numerical experiment of the paper.

**Methods And Evaluation Criteria:**

The method makes sense but the numerical experiment is lacking.

**Other Comments Or Suggestions:**

Please refer to the questions below.

**Other Strengths And Weaknesses:**

Strength:

The paper develops the clever idea of considering the augmented state space by letting a state variable to denote the budget. The paper achieves the first sublinear regret for this type of risk-sentitive CMDP.

Weakness:

1. The current method applies to the case with a single-constraint. Though the authors claim that there is a way to extend to multiple constraints, it is not immediately clear to me and I suspect the current approach would make the computation time exponential in the number of constraints. Please refer to the questions below for more details over this weakness.

2. It is not clear how the developed algorithm works in practice due to the lack of numerical experiments.

3. The computation complexity of their algorithm has not been discussed.

**Questions For Authors:**

1. It is discussed in the extension that the current approach could extend to the multiple constraint case. However, it seems that the current approach optimizes over the augmented variable $\tau$ by discretizing the range. If there are multiple constraints, will it be the case that a multi-dimensional space would need to be discretized? This will make the computation time exponentially large.

2. Is there any intuition on the lower bound of the regrets for the risk-sensitive CMDP?

3. Could you please provide any comment on the practical performance of the algorithm developed in the paper?

**Relation To Broader Scientific Literature:**

The paper develops the first sublinear regret for the risk-sensitive CMDP.

**Theoretical Claims:**

The claims and proofs look correct to me.

---

> ### Author Rebuttal · Authors · 2025-04-01
>
> We thank the reviewer for providing thoughtful comments. Please see our responses to your questions below.
>
> >*For multiple constraints*
> * As correctly pointed out by the reviewer, the complexity of our approach is affected by the number of constraints. Specifically, extending the method proposed in the paper to multiple constraints requires augmenting the state with multiple budget variables, which leads to an exponential increase in the state space. For $M$ constraints, the size of the augmented state space scales as $ (\frac{C}{\epsilon})^M $, where  $C$ is the upper bound of the augmented budget $\tau$. We thank the reviewer for highlighting this issue, and we will update the statement in the paper accordingly.
>
> >*Lack of numerical experiments*
> * Our main contribution is theoretical in nature. To the best of our knowledge, this is the first work that establishes sublinear regret and constraint violation bounds in the setting where the goal is to maximize the expected cumulative reward subject to an entropic risk constraint on the utility.
> * To address this, we leverage the augmented MDP representation of OCE-based risk measures, which includes the entropic risk measure as a special case. This idea originates from [Bäuerle and Ott, 2011], where the authors studied the memory requirements for solving CVAR in the absence of a Bellman equation, and was further developed in [Wang et al., 2024] to reduce OCE problems to standard reinforcement learning.
> * Our work is the first to tackle the CMDP setting with an entropic risk constraint, using the augmented MDP framework to address the nonlinearity in optimizing the Lagrangian, and defining a composite value function.
> * Note that even though we use an augmented budget, we do not need to augment all the history, rather, we only need to augment the remaining budget from the initial value. Also, since we assume the utilities are deterministic, the transition of the augmented budget is known. Thus, we expect that our approach can be implemented in practice. We will add numerical results in the final version of the paper.
>
> >*Computational Complexity*
>
> * The computational complexity is O(K), i.e. a linear in $K$, and thus, our approach is computationally efficient. In particular, if we discretize $\tau$ with spacing $1/K$, then this will add an additional $H/K$ factor in the regret bound (as the OCE representation for entropic risk measure is 1-Lipschitz) over every episode that results in $O(H)$ regret overall which is independent of $K$. Thus the maximization over $\tau$ can be done in $O(K)$ time, which is linear with $K$, hence, efficient. We have discussed the computational complexity aspect in Section 4 (please see the paragraph just before Section 5). Note that the augmented budget is also considered in the unconstrained entropic risk or CVaR maximization problem [Wang et al.'2023,2024]. They also discretize the augmented budget and the computational complexity is also of the same order as ours.
>
> >*Lower Bound*
> * Regarding regret lower bounds for CMDP, there are currently no known results for cumulative reward and utility, making this a largely unexplored area. The presence of an entropic constraint further complicates the analysis due to the absence of strong duality.
>
> >*Regarding the references*
>
> * Thanks for pointing out the references.  These are really interesting, and we will include them in the final version. In the following, we recap our contributions to the other works. Note that to the best of our knowledge, this is the first work that achieves sublinear regret ($O(\sqrt{K})$) and sublinear violation bound ($O(K^{3/4})$) for risk-constrained RL problem. Chow et al.'2018 did not consider the regret and violation bound.  In order to achieve this, we contribute significantly. We have to use a regularized primal-dual approach, unlike the risk-neutral CMDP setting, as the strong duality does not hold. Further, we cannot apply dynamic programming-based approach to the composite state-action value function because of the non-linearity of the value function with respect to the state-action occupancy measure, and we have to resort to the OCE representation. In the OCE representation, we have to augment the state with a budget, and then we optimize over the budget.
> * Some studies also use an LP-based approach to bound the regret and the violation in the risk-neutral setting. They use the state-action-occupancy measure rather than a policy to optimize. However, the state-action occupancy-based measure (and thus, the LP-based approach) will not work for our problem as the value function is not linear in terms of the state-action occupancy measure.

---

### Official Review · Reviewer_c96z · 2025-03-14

**Overall Recommendation:** 3

**Summary:**

The paper studies online learning in episodic finite-horizon constrained Markov Decision Processes  with entropic risk-sensitive constraints. Traditional primal-dual methods fail to directly address risk-sensitive CMDPs due to the non-linear nature of entropic risk constraints and the lack of strong duality. The authors propose augmenting the CMDP by incorporating a budget variable into the state-space, allowing the use of value iteration. They then introduce a primal-dual algorithm with regularized dual updates and prove the first known sublinear regret and violation bounds  for risk-sensitive CMDPs.

**Claims And Evidence:**

- The major claims of achieving sublinear regret and violation bounds for the risk-sensitive CMDP setup are well-supported by theoretical arguments and detailed proofs.
- The paper claims that the augmented CMDP allows for a tractable solution to the risk-sensitive constraint, but the paper does not provide a computational complexity analysis.

**Essential References Not Discussed:**

No

**Experimental Designs Or Analyses:**

The paper is purely theoretical, and no empirical experiments are presented. While  its absence does not detract from the theoretical contributions, numerical validation could strengthen the paper.

**Methods And Evaluation Criteria:**

The introduction of an augmented CMDP framework to overcome the challenge of nonlinear entropic risk measures is inspired by (B¨auerle & Ott, 2011; Wang et al., 2024), which is sensible. While the evaluation criteria, specifically regret and constraint violation, seems quite standard for the considered setting, I do have some questions regarding the learning metric. Does the metric is defined over the admissible/feasible policy set? If not, then some policy may achieve negative violation. Should we take the positive part of violation then?
 Another concern regarding evaluation is that the paper lacks empirical evaluation, including simple numerical experiements.

**Other Comments Or Suggestions:**

- A more detailed discussion on why entropic risk measures, rather than CVaR or other risk measures, are particularly valuable would help clarify the practical implications. In contrast, CVaR or VaR-constrained  formulation is well-known and well motivated

- There are also some typos throughout the paper. Need proofreading.

**Other Strengths And Weaknesses:**

Strengths:
- First rigorous theoretical analysis of CMDPs with entropic risk-sensitive constraints.
- Clear theoretical exposition and solid analytical framework.


Weaknesses:
- Lack of numerical examples or empirical validation
- The dual regularization strategy is somewhat heuristic; further justification or exploration of alternatives would strengthen the approach.

**Questions For Authors:**

- Can the authors discuss the implications of regularization parameter  in more detail, including its choice and sensitivity?

- Regarding the regret bound in section 5.1, the lower bound result for ERM-MDP is improved and proven tight in Liang and Luo 2024, in contrast to Fei et al. 2021, which is not discussed. While it might be challenging, is it possible to derive or adapt lower bounds to this setting? How tight are your proposed upper bounds likely to be compared to these potential lower bounds?

- The author should also carefully discuss the worsening of bounds compared with risk-neutral CMDP in terms of SAKH, rather than K only.

- Following the idea of (Wang et al., 2023,2024), it seems to be possible to directly extend the whole framework from ERM-constrained setting to OCE-constrained setting.

Fei, Y., Yang, Z., Chen, Y., and Wang, Z. Exponential bellman equation and improved regret bounds for risksensitive reinforcement learning. In Proceedings of the 35th International Conference on Neural Information Processing Systems, NIPS ’21

Wang, K., Kallus, N., and Sun, W. Near-minimax-optimal risk-sensitive reinforcement learning with cvar. In International Conference on Machine Learning, pp. 3586435907. PMLR, 2023.

Wang, K., Liang, D., Kallus, N., and Sun, W. Risk-sensitive rl with optimized certainty equivalents via reduction to standard rl. arXiv preprint arXiv:2403.06323, 2024.

Liang, H., & Luo, Z. Q. (2024). Bridging distributional and risk-sensitive reinforcement learning with provable regret bounds. Journal of Machine Learning Research, 25(221), 1-56.

**Relation To Broader Scientific Literature:**

The paper  situates its contribution within the existing literature on CMDPs and risk-sensitive RL. It  dentifies gaps related to nonlinear risk constraints and appropriately builds upon recent advances (Wang et al., 2023; Ding & Lavaei, 2023) and extends primal-dual methods but adapts them to risk-sensitive settings.

**Theoretical Claims:**

I checked the main theoretical claims, specifically the correctness of Lemmas 3.1 and 4.1, which  the value-function equivalences and existence of Markovian optimal policies. No significant issues were identified; the derivations appear mathematically sound and clearly presented.

---

> ### Author Rebuttal · Authors · 2025-04-01
>
> We thank the reviewer for providing thoughtful comments. Please see our responses below.
> >*Lack of numerical results*
> * Our main contribution is theoretical. To the best of our knowledge, this is the first work that establishes sublinear regret and constraint violation bounds in the setting where the goal is to maximize the expected cumulative reward subject to an entropic risk constraint on the utility.
> * To address this, we leverage the augmented MDP representation of OCE-based risk measures, which includes the entropic risk measure as a special case. This idea originates from [Bäuerle and Ott, 2011], where the authors studied the memory requirements for solving CVAR in the absence of a Bellman equation, and was further developed in [Wang et al., 2024] to reduce OCE problems to standard reinforcement learning. The OCE representation is necessary since the standard dynamic-programming-based approach does not apply to the composite state-action value function. We will add numerical results in the final paper in Appendix.
>
> >*Dual regularization..*
> * Minimizing the regularized Lagrangian $V_r^{\pi} + \lambda (V_g^{\pi} - b) + \beta \lambda^2$ with respect to $\lambda$ yields $V_r^{\pi} - \frac{1}{4\beta}(V_g^{\pi} - b)^2$ if $V_g^{\pi} - b < 0$, and $V_r^{\pi}$ otherwise. Thus, adding the regularization is similar to minimizing the $\ell_2$ loss of the constraint violation, which facilitates bounding the violation term. In particular, we show that this regularization leads to an $O(K^{3/4})$ bound on the cumulative violation.
>
> >*Choice of ERM rather than CVaR..*
>
> The entropic risk measure prioritizes policies with desirable robustness and performance trade-offs. Specifically:
> * **Robustness:** The entropic risk measure is connected to robustness. It has been shown that maximizing the entropic risk measure with a risk sensitivity parameter $\alpha$ is equivalent to optimizing the worst-case expected return under a distributional ambiguity. More precisely, this corresponds to maximizing the minimum performance over a set of distributions within a KL ball of radius $\alpha$ around the nominal distribution.
>
> * **Dynamic Programming:** Unlike CVaR or VaR, the entropic risk measure is smooth and satisfies a multiplicative form of the Bellman equation. This makes it particularly amenable to gradient-based optimization algorithms.
>
> * **Sensitivity to the Risk Parameter:** As the risk factor $\alpha$ approaches infinity, the entropic risk measure increasingly emphasizes adverse outcomes. However, unlike CVaR, which explicitly focuses on tail risk, it still retains sensitivity to the entire distribution of outcomes.
>
> >*Implications of the regularization term*
> * The added regularization ensures the boundedness of the violation, which is essential for our analysis and the derivation of both the constraint violation and the regret bounds. As the regularization parameter increases, the algorithm increasingly prioritizes smaller values of $\lambda$, eventually rendering the resulting algorithm ineffective. Therefore, a small choice of $\beta$ is necessary for maintaining good performance. Specifically, as stated in Lemma 5.2, there exists a trade-off between the step-size and the regularization parameter, i.e., $\beta \eta$ must be smaller than 0.5.
>
> >*Regarding lower bound*
> * We thank the reviewer for mentioning the work of [Liang and Luo, 24]. Extending the results of [Liang and Luo, 2024] would be quite challenging, but represents an interesting future direction. Specifically, the distributional RL framework for entropic risk follows a similar multiplicative Poisson equation as the one we study and thus encounters similar challenges to those discussed in our paper.
> * Regarding regret lower bounds for CMDP, there are currently no known results, making this a largely unexplored area. The presence of an entropic constraint further complicates the analysis due to the absence of strong duality.  Nonetheless, one might be able to extend the general ideas from [Liang and Luo, 2024] by adopting an augmented MDP approach instead. We agree this is a promising direction and have added a sentence discussing it in the paper. Our regret bound is the same as that of the risk-neutral CMDP.
>
> >*Extending other OCE representations*
> * The main focus of this paper has been to extend the CMDP framework to the entropic risk minimization setting. To tackle this problem, we leverage the augmented MDP representation introduced by [Wang et al., 2023, 2024]. This approach, however, comes at the cost of discretizing the parameter $\tau$, for which we currently do not have a complete solution. We also acknowledge that the proposed framework can be generalized to a broader class of risk measures that admit an OCE representation. This extension can build upon the ideas of [Wang et al., 2024] and our current analysis. However, such a generalization is non-trivial and involves additional technical challenges that constitute future research direction.

---

> > ### Comment · Reviewer_c96z · 2025-04-04
> >
> > Thank the authors for their detailed response and clarifications, which address most of my concern. I maintain my positive evaluation for this paper.

---

> > > ### Author Response · Authors · 2025-04-05
> > >
> > > We are glad that our responses have clarified most of your concern, and would like to thank you for your support. Could you please consider raising the score?

---

### Official Review · Reviewer_DeSK · 2025-03-14

**Overall Recommendation:** 2

**Summary:**

This work focuses on Risk-Sensitive Constrained MDPs and proposes a novel algorithm that ensures the entropic risk for an additional utility function remains above a given threshold. Under this setting, the author introduces a new algorithm based on the primal-dual method for an augmented MDP. Additionally, the author provides the first sub-linear regret guarantee for the reward in this setting, along with a non-optimal violation magnitude for the constraints.

**Claims And Evidence:**

The author provides a clear claim of the result in the theorems and includes a proof sketch to outline the key steps of the theoretical analysis.

**Essential References Not Discussed:**

This paper provides a comprehensive discussion of related work in risk-sensitive reinforcement learning and constrained reinforcement learning.

**Experimental Designs Or Analyses:**

The main contribution of this work focuses on the theoretical analysis of regret and does not have experiment.

**Methods And Evaluation Criteria:**

The main contribution of this work focuses on the theoretical analysis of regret and does not have experiment.

**Other Comments Or Suggestions:**

1. It is not clear why violation is an appropriate measure for constraint RL. Specifically, in a strictly constrained setting, the utility should be above the threshold in each episode, rather than being averaged over multiple rounds. Even in a soft-constrained setting (such as in classification problems), a more natural approach would be to use a truncated violation measure like $\max(0, B - V_g)$ in each round, preventing a scenario where a round with high utility compensates for rounds with low utility. Furthermore, even if such compensation is allowed, given that the author considers a risk-sensitive utility function, it would be more reasonable to introduce an entropy-based loss structure for the compensation process rather than using a linear summation, which is more suitable for risk-neutral cases.

2. In this work, the author considers a deterministic reward. In the traditional RL framework, it is natural to transform the reward into its expectation, and unbiased noise does not significantly affect the learning process, as most challenges arise from learning the transition dynamics. However, it is unclear whether the assumption of a deterministic reward is reasonable in a risk-sensitive environment. Will this assumption have further implications for the entropy-type value function?

**Other Strengths And Weaknesses:**

1. The proposed algorithm is highly computationally inefficient due to the augmented feature of the budget variable. Specifically, in tabular MDPs, the number of state-action pairs is finite, allowing for efficient value function updates. However, the augmented budget variable can take any real value, making the value function update (Lines 13-14) computationally expensive, as it must be performed over a continuous range of budget values. Additionally, the optimization of $\tau$ in Line 16 may also be inefficient, especially if the value function lacks useful properties such as convexity. A discussion on potential computational improvements or approximations would be beneficial.

2.The algorithm appears to be a direct combination of the existing primal-dual method used in constrained RL and the augmented method used in risk-sensitive RL. As a result, the novelty of the approach is limited.

3. It is not clear why violation is an appropriate measure for constraint RL. Specifically, in a strictly constrained setting, the utility should be above the threshold in each episode, rather than being averaged over multiple rounds. Even in a soft-constrained setting (such as in classification problems), a more natural approach would be to use a truncated violation measure like $\max(0, B - V_g)$ in each round, preventing a scenario where a round with high utility compensates for rounds with low utility. Furthermore, even if such compensation is allowed, given that the author considers a risk-sensitive utility function, it would be more reasonable to introduce an entropy-based loss structure for the compensation process rather than using a linear summation, which is more suitable for risk-neutral cases.

**Questions For Authors:**

1.In the learning matrix, why consider a linear summation for the compensation process rather than an entropy-based loss structure, which is more aligned with risk-sensitive settings?

2. 1. deterministic reward

**Relation To Broader Scientific Literature:**

This work mainly focuses on risk-sensitive constrained reinforcement learning; however, the proposed algorithm is highly computationally inefficient, making it primarily relevant for the theoretical analysis of reinforcement learning rather than practical applications.

**Theoretical Claims:**

The novel algorithm combines the augmented method with the primal-dual method, where the augmented method is widely used in risk-sensitive reinforcement learning, and the primal-dual approach is commonly applied in constrained MDPs. Therefore, the algorithm is well-motivated. The author provides a clear proof sketch, and there are no concerns about correctness based on the proof sketch.

---

> ### Author Rebuttal · Authors · 2025-04-01
>
> Thanks for providing thoughtful comments. Please see our responses below.
> >*Regarding the computational efficiency...*
> * *Even though the augmented budget can take value in real space, the computational complexity is still O(K), i.e. a linear in K. Thus, it is **not true** that our approach is computationally inefficient.* In particular, if we discretize $\tau$ with spacing $1/K$, then this will add $H/K$ factor in the regret bound (as the optimized certainty equivalent (OCE) representation for entropic risk measure is 1-Lipschitz) over every episode that results in $O(H)$ additional regret overall which is independent of $K$. Thus the maximization over $\tau$ can be done in $O(K)$ time which is linear with $K$, hence, efficient. We have discussed the computational complexity aspect in Section 4 (please see the paragraph just above Section 5).  Note that the augmented budget is also considered in the unconstrained entropic risk or CVaR maximization problem [Wang et al'23], and the computational complexity is also of the same order as ours.
> * While the augmented budget can take real value, the constrained entropic risk-measure problem is inherently challenging. *In particular, one cannot apply the dynamic programming-based approach to the Lagrangian or the composite state-action value function, unlike the risk-neutral CMDP approach.* Note that unconstrained entropic risk-sensitive RL admits an optimal Bellman equation and one can directly apply the dynamic programming-based approach there without augmentation. However, we cannot extend those approaches here. Hence, we need to resort to the OCE representation of the entropic risk measure. As a result, we augment the state space with the budget and then solve for the optimal budget.
>
> >*..the novelty of the approach is limited.*
>
> In the following, we state the novelty of our proposed approach.
> * Our study shows that in the constrained entropic risk-sensitive RL problem one cannot apply the dynamic programming-based approach on the composite state-action value function, unlike the risk-neutral CMDP approach. Hence, we need to resort to the OCE representation of the entropic risk measure. The key here is that we can write the Bellman equation there for the composite state-action value function in the OCE representation. Hence, one can obtain an efficient computation approach.
> * The additional challenge comes from the fact that the entropic risk measure is not linear in the state-action occupancy measure even with the augmented state-spaced. Hence, the traditional primal-dual-based approach cannot be applicable to bound the violation, unlike the risk-neutral CMDP setup. We resort to the regularized primal-dual-based approach.
> * Further, for the unconstrained augmented MDP problem, one uses the fact that the greedy policy is optimal with respect to the augmented problem to achieve the bound. However, in the constrained setting, the greedy policy with respect to the composite state-action value function (in the augmented state space) might not be feasible and, hence, might not be optimal. Hence, we have to use novel proof techniques.
> * Overall, to the best of our knowledge, this is the first result that shows that $O(\sqrt{K})$ (sublinear) regret and the $O(K^{¾})$ (sublinear) violation bound is achievable in the constrained risk-sensitive setting. In order to achieve this, we have to identify and apply tools in a novel manner and this will influence many new approaches in the future.
>
> >*Other violation metric..*
> * The violation metric we consider is common in the risk-neutral CMDP literature. Note that even for this violation metric, the traditional primal-dual-based approach that can bound the violation for the risk-neutral setting is not applicable here  since the strong duality may not hold. Hence, bounding this violation metric is challenging in our setting.
> * We agree with the reviewer that the truncated violation like $\max(0,B-V_g)$ might be a better alternative for online learning. However, how to achieve optimal regret along with the truncated violation in a computationally efficient manner using a primal-dual approach still remains open *even in the risk-neutral CMDP*.  The recent work [A1] achieves $O(\sqrt{K})$ regret and $O(\sqrt{K})$ *truncated* violation in the risk-neutral CMDP using a double loop technique. However, the computational complexity of the proposed approach is exponential in terms of $H$ and $K$. We can use a similar technique to bound the truncated violation in our setting. However, the computational complexity will still be exponential. We will mention the above in the final version. Because of the above reason, we did not explicitly consider the truncated violation metric. How to develop a computationally efficient approach to bound both the regret and the truncated violation has been left for the future.
>
> [A1]. Ghosh et al. "Towards achieving sub-linear regret and hard constraint violation in model-free rl." AISTATS,2024.

---

> > ### Comment · Reviewer_DeSK · 2025-04-03
> >
> > Thanks for the rebuttal. I still have a concern regarding the violation metric. According to the authors, it seems acceptable to allow non-truncated violations in each round. However, as I mentioned in the “Weaknesses and Questions” section, a main concern remains: why is the violation metric defined as the summation of violations across stages?
> >
> > While linear summation is standard in risk-neutral environments for both rewards and violations, it may not be appropriate in risk-sensitive settings. For example, a risk-sensitive user would clearly prefer a trajectory with violations of (0, 0) over (1, -1), even though their sums are equal. This suggests the current metric may fail to capture meaningful distinctions in risk-sensitive scenarios.
> >
> > Furthermore, even if some form of compensation across rounds is allowed, using a linear summation seems misaligned with the stated risk-sensitive objective. It would be more appropriate to incorporate an entropy-based or utility-weighted loss structure that better reflects risk sensitivity.
> >
> > Therefore, I will keep my score unchanged.

---

> > > ### Author Response · Authors · 2025-04-03
> > >
> > > Thanks for your comments. In the following, we address your comment.
> > >
> > > >*..a main concern remains: why is the violation metric defined as the summation of violations across stages?..*
> > >
> > > We believe the reviewer may be conflating two related but distinct notions: *constraint violation* and the *risk-sensitive nature of the constraint*. We'd like to emphasize this distinction.
> > >
> > > The *risk sensitivity* in our setting is captured through the entropic risk associated with the utility function.  In contrast, the *constraint* imposes a hard threshold that defines the admissible set of policies, within which we aim to find the optimal one. Importantly, under this formulation, the agent inherently prefers not to violate the constraint at all. Even when considering cumulative utility, the agent would favor $(0, 0)$ over $(1, -1)$, as the latter involves a violation, whereas the former does not. Moreover, the agent has no preference between $(0, 10)$ and $(5, 5)$, as both satisfy the constraint and do not result in any violation.
> > >
> > > While ideally we would like to measure the total number of time steps at which constraint violations occur (i.e., a binary indicator per round), this formulation is difficult to analyze and optimize over. The next-best alternative is the *truncated violation*, which does reflect constraint satisfaction more faithfully but introduces significant nonlinearity and is also analytically challenging.
> > >
> > > As we mentioned in our earlier rebuttal, how to minimize the truncated violation metric $\max(0, B - V_g)$ using a computationally efficient approach is still an open question, even for risk-neutral MDPs. Hence, the consideration of such a metric is beyond the scope of this paper. **Instead, we show that although dynamic programming-based approaches and Markovian policies are optimal in the unconstrained case, they are no longer applicable in the constrained case; even when considering the composite state-action value function.** *To address this, we leverage the Optimized Certainty Equivalent (OCE) representation and demonstrate how state augmentation can be used to minimize both regret and violation.*
> > >
> > > Furthermore, standard primal-dual methods cannot be directly applied here since strong duality may fail to hold; the value function is no longer linear in the state-action value function. To overcome this, we introduce a regularization term and derive bounds for both regret and violation. We believe these insights can serve as a foundation for future work that explores alternative violation metrics in risk-sensitive settings.
> > >
> > > As a practical compromise in this work, we adopt the *linear (untruncated) violation metric*, which, though weaker, has been widely used and empirically shown in many risk-neutral settings to correlate well with truncated violations. Note that here, the violation means how much the entropic risk measure associated with the policy violates from $B$, here, it does not consider the realized value as the reviewer might be suggesting.  Moreover, constraint satisfaction can often be improved by slightly tightening the constraint threshold, e.g., by using $B + \epsilon$ instead of $B$. This approach can yield strong empirical guarantees without sacrificing regret bounds in risk-neutral MDPs, and we believe similar techniques can be extended to the risk-sensitive setting as well.
> > >
> > > We hope this clarification addresses the reviewer's concerns regarding the metric’s role and justification within our framework.

---

### Decision · Program_Chairs · 2025-05-01

**Decision:**

Accept (poster)

**Comment:**

The paper makes a clear and novel contribution by proposing an algorithm for a specific class of risk-sensitive constrained Markov Decision Processes and establishing both sublinear regret and a bound on constraint violation--results that are the first for this particular formulation. The analysis leverages nontrivial techniques such as state augmentation, and no major flaws were identified during the review process.

However, the practical relevance of the work remains uncertain. This is partly due to the absence of numerical experiments, which limits the reader's ability to assess the empirical behavior of the proposed approach. Given that the studied formulation appears to be chosen in part for analytical tractability--rather than being the most natural or widely applicable model--it is especially important for the authors to include the promised experimental results in the final version.